# Efficient Skill Grounding via Code Refactoring with Small Language Models

**Sera Choi** [1] **Wonje Choi** [1] **Saehun Chun** [1] **Daehee Lee** [1] **Jooyoung Kim** [1] **Chaeun Lee** [2] **Honguk Woo** [1]

## Abstract

Effective skill grounding is essential for deploying reusable skills in embodied agents, as even minor embodiment or environmental differences can render an entire skill incompatible. This challenge is particularly pronounced in embodied settings, where agents must operate in dynamic, partially observable environments without access to large language models (LLMs). In this setting, reliance on LLMs is impractical, while small language models (sLMs) remain insufficient for the effective skill grounding required for reliable long-horizon control. We present RECENT, a refactoring-centric agent framework that enables efficient skill grounding with sLMs by decoupling skill semantics from embodiment- and environment-specific execution binding. By representing skills as executable code, RECENT preserves the semantic intent encoded in a skill's control structure while grounding it by modifying only execution bindings through localized refactoring, rather than regenerating code from scratch. We evaluate RECENT across diverse skill grounding scenarios spanning multiple robot embodiments in dynamic environments, demonstrating robust long-horizon performance when deployed with an sLM. Across all scenarios, RECENT achieves the best performance among sLM-based Code-as-Policies (CaP) methods and matches the task performance of LLM-based CaP.

## 1. Introduction

Recent embodied control systems increasingly exploit the planning capabilities of large language models (LLMs) to solve tasks by composing learned skills represented as neural sub-policies or code snippets (Brohan et al., 2023; Wang et al., 2023a). A skill inherently couples functional semantics, which specify what is to be achieved, with executable components, which determine how the skill can be realized by a particular robot in a given environment. However, because robots differ in morphology, actuation, and sensing, and task environments vary in object properties, physical constraints, and operational conditions, the executability of a skill is highly deployment-dependent, rendering direct skill reuse across contexts infeasible. Consequently, existing skill representations make it difficult to explicitly separate functional semantics from executable components, hindering deployment-time grounding and leading to relearning or regeneration when execution contexts change.

Such designs incur substantial computational overhead, frequently requiring additional training to relearn neural sub-policies when deployment conditions change. Representing skills as executable code partially alleviates this issue by enabling grounding through inference at test time rather than retraining. However, existing code-based approaches still typically rely on regenerating the entire skill implementation, instead of adapting only deployment-specific components while preserving the overall skill structure. This regeneration overhead becomes particularly pronounced on capacity-limited devices, where online access to large-scale LLMs is not guaranteed and skill grounding must be performed using on-device computing resources, rendering LLM inference impractical. On the other hand, small language models (sLMs) enable efficient inference but offer limited reasoning capacity (Choi et al., 2024), and existing skill grounding approaches that rely on regeneration remain ill-suited for dynamic environments.

To address these challenges, we present RECENT, a refactoring-centric agent framework that enables sLMs to perform efficient skill grounding. Skill grounding becomes efficient when invariant semantic intent is separated from deployment-specific execution bindings, allowing grounding to be bootstrapped across diverse deployment contexts. By representing skills as well-defined executable code, RECENT preserves semantic intent in functional logic that can be reused across embodiment and environment differences, while isolating execution bindings as localized components for on-demand modification. As a result, sLMs are restricted to localized editing at deployment time, resolving

[1]Department of Computer Science and Engineering, Sungkyunkwan University, Suwon, Republic of Korea [2]Department of Systems Management Engineering, Sungkyunkwan University, Suwon, Republic of Korea. Correspondence to: Honguk Woo <hwoo@skku.edu>.

*Proceedings of the 43$^{rd}$ International Conference on Machine Learning*, Seoul, South Korea. PMLR 306, 2026. Copyright 2026 by the author(s).

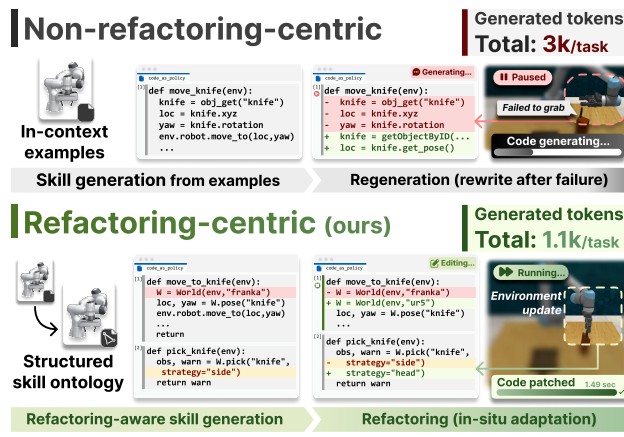

*Figure 1.* Key concept comparing (*top*) the skill grounding procedure used in existing approaches with (*bottom*) our refactoring-centric skill grounding procedure.

embodiment mismatches through lightweight code modifications without the extensive reasoning associated with regenerating code from scratch. Environmental variations are handled through in-situ adaptation, where execution-time feedback is incorporated to progressively patch the executing code without frequent execution interruptions. Figure 1 illustrates a code-level comparison between existing skill grounding approaches that rely on inefficient full regeneration and our framework, which achieves efficient skill grounding through localized code refactoring.

Specifically, we adopt a skill ontology to declaratively encode skill semantics, robot capabilities, and their relationships, providing a unified foundation for reusable skill representations. Guided by this ontology, we construct (i) an offline skill repository using an LLM, in which semantic intent is explicitly encoded in executable skill code along with metadata describing functional requirements and adaptation cues. By validating this semantic intent offline on a common robot platform prior to deployment, subsequent skill grounding no longer needs to reason about complex task semantics and can instead focus solely on resolving execution bindings. Deployment-time skill grounding in RECENT addresses embodiment mismatches through (ii) ontology-based reasoning and environmental variations through (iii) in-situ adaptation, both realized via Fill-in-the-Middle (FIM) (Bavarian et al., 2022)-based localized code refactoring rather than end-to-end regeneration from scratch. Embodiment mismatches are amenable to sLM-based editing because relevant execution bindings are explicitly identified through ontology-level comparisons between skill requirements and target robot capabilities, constraining grounding to localized pre-execution code edits. In contrast, environmental variations are handled by deferring adaptation to execution time, during which an sLM proactively patches yet-to-be-

executed code snippets under unit-level validity checks, preserving executability without altering global task semantics.

We evaluate RECENT across diverse skill grounding scenarios using sLMs under deployment constraints, spanning multiple robot embodiments, dynamic environments, and capacity-limited device settings. Specifically, we design diverse long-horizon robotic manipulation tasks spanning multiple robot embodiments in CoppeliaSim (Rohmer et al., 2013) and Genesis (Genesis Authors, 2024). Across all scenarios, RECENT, deployed with the sLM Qwen2.5-Coder-7B (Hui et al., 2024), outperforms the Code as Policies (CaP) baseline (Liang et al., 2023) instantiated with the same-sized distilled sLM CodeV-R1 (Zhu et al., 2025), achieving an improvement of 58.81 percentage points (pp) in task success rate (SR), a 99.09 pp reduction in grounding overhead (GO), an average of only 0.71 execution interruptions (EI) per task, approaching zero, and a 93.29% relative reduction in idle time (IT). Its performance is comparable to that of LLM-based CaP using GPT-5.2-Codex (OpenAI, 2025), with differences of only 6.58 pp in SR, while outperforming it on average across the remaining metrics despite operating under deployment constraints, as shown in Table 1, achieving a substantial 57.81 pp improvement in GO and average reductions of 22.95% and 77.52% in EI and IT, respectively.

Our contributions are summarized as follows:

- We present RECENT, a refactoring-centric agent framework that enables efficient skill grounding with sLMs under deployment constraints, making long-horizon control practical without reliance on large-scale LLM inference.

- We represent skills as executable code that separates invariant semantic intent from deployment-specific execution binding, enabling skill grounding through localized code refactoring rather than regeneration from scratch.

- We evaluate RECENT on diverse skill grounding scenarios and show consistent performance improvements over distilled sLM-based CaP in SR, GO, EI, and IT, while remaining comparable to LLM-based CaP.

## 2. Related Work

**LLM-based Embodied Control.** In embodied control, recent work increasingly leverages the reasoning capabilities of LLMs for high-level task planning over predefined skill policies (Huang et al., 2022; Brohan et al., 2023; Song et al., 2023). Building on recent advances in the code-writing capabilities of LLMs (Chen et al., 2021; Roziere et al., 2023; Hui et al., 2024; Guo et al., 2024; Zhu et al., 2024) embodied policies can be represented and executed in programmatic forms, commonly referred to as *Code-as-Policies* (Liang et al., 2023; Huang et al., 2023b;a; Burns et al., 2024; Li et al., 2024; Mu et al., 2024; Vemprala et al., 2023; Singh et al., 2022; Wang et al., 2023b). Rather than mapping in-

structions to a fixed set of predefined skills, these methods prompt LLMs to generate Python-like programs that directly invoke perception and motor APIs, enabling motion-level control in embodied agents. Unlike existing approaches that rely on large-scale LLMs at deployment, our work focuses on reliable long-horizon embodied control under deployment constraints by representing skills as reusable code and enabling sLMs to ground them through localized code refactoring rather than regeneration from scratch.

**Skill Grounding.** In embodied agents, skills are temporally extended and reusable behavioral patterns that encapsulate low-level control, enabling higher-level planning and composition for solving complex tasks (Kober & Peters, 2009; Rozo et al., 2020; Kroemer et al., 2021). Parametric approaches typically ground such skills through retraining or fine-tuning neural policies, mapping abstract skill representations to concrete robot actions while entangling task semantics with execution details within the learned policies (Xu et al., 2023; Wang et al., 2024; Doshi et al., 2024). More recently, LLM-based embodied agents have represented skills as function-level code, where task-level decisions and execution-specific details are generated jointly within a single program (Tziafas & Kasaei, 2024; Sarch et al., 2023; Li et al., 2025a). We focus on how skills should be structured to support efficient grounding. Specifically, we separate semantic intent from execution context, so that functional logic is preserved, while only execution-specific components are subject to adaptation. This separation reduces the reasoning burden on the sLM by preserving semantic intent in functional logic and restricting deployment-time grounding to localized edits over execution bindings. As a result, skill grounding naturally reduces to refactoring execution-specific code fragments rather than regenerating entire skill implementations from scratch.

**Programmatic Control.** Recent advances in LLMs have spurred growing interest in programmatic control, where agents generate, execute, and repair code to solve complex tasks (Xia & Zhang, 2023; Yang et al., 2025; Bouzenia et al., 2025; Xia et al., 2025). By explicitly exposing control flow and intermediate program structure, these approaches facilitate structured reasoning, improved generalization, and compositionality compared to end-to-end learned control policies. Programmatic control has also been applied to embodied agents, enabling code generation and execution over perception and control modules (Liang et al., 2023; Huang et al., 2023b). Unlike programmatic agents in digital environments, embodied code agents must reason over continuous states and interact with the physical world under partial observability (Ahn et al., 2025; Meng et al., 2025; Ying et al., 2025). In embodied settings, generated code often coordinates perception, decision-making, and actuation in a closed-loop manner, where continuous control

and environmental feedback are essential for handling uncertainty and long-horizon dependencies. Unlike existing approaches, RECENT performs in-situ adaptation to environmental variations during execution, enabling continuous control without interrupting program execution.

## 3. Problem Formulation

We formulate an embodied task as $\tau = (\mathcal{S}, \mathcal{A}, \mathcal{G}, T)$, where $\mathcal{S}$ and $\mathcal{A}$ denote the state and action spaces, respectively. Due to partial observability (Sutton & Barto, 2018), at each timestep $t$ the agent receives an observation $o_t$ that provides incomplete information about the underlying state $s_t \in \mathcal{S}$. The environment dynamics are defined by the transition function $T : \mathcal{S} \times \mathcal{A} \to \mathcal{S}$. We denote by $\mathcal{G} \subset \mathcal{S}$ the set of goal states, and define each task $\tau$ as a composite objective consisting of multiple individual goals $g \in \mathcal{G}$. To solve a task $\tau$, the agent is provided offline with a set of reference skills $\mathcal{X} = \{\chi_1, \ldots, \chi_K\}$, where each skill $\chi_k$ is a temporally extended action and may not be directly executable in the target deployment setting. Our objective is to optimize an sLM $\pi_{\text{sLM}}$ over a set of tasks $\mathcal{T}$ to maximize task success rates while minimizing skill grounding overhead and preserving execution continuity:

$$
\max_{\pi_{\text{sLM}}} \mathbb{E}_{\tau \sim \mathcal{T}} \Big[ \text{SR}(\pi_{\text{sLM}}, \tau) \\
- \lambda \text{C}_{\text{gro}}(\pi_{\text{sLM}}, \tau; \mathcal{X}) - \mu \text{C}_{\text{exe}}(\pi_{\text{sLM}}, \tau) \Big]. \tag{1}
$$

Here, $\text{SR}(\pi_{\text{sLM}}, \tau)$ denotes the task success rate for executing $\tau$ under the decisions of $\pi_{\text{sLM}}$, indicating whether all goal conditions are satisfied. The grounding cost $\text{C}_{\text{gro}}(\pi_{\text{sLM}}, \tau; \mathcal{X})$ measures the deployment-time overhead incurred when grounding $\chi_k \in \mathcal{X}$ to $\tau$. The execution cost $\text{C}_{\text{exe}}(\pi_{\text{sLM}}, \tau)$ penalizes execution disruptions that interrupt task progress. The coefficients $\lambda$ and $\mu$ balance task success against grounding and execution costs.

## 4. Approach

We present RECENT, a code refactoring-centric agent framework that enables sLMs to perform efficient skill grounding under deployment constraints, ensuring task success, minimizing grounding overhead, and preserving execution continuity. In RECENT, skills are generated and validated by an LLM with a clear separation between semantic intent and execution-specific components within the skill code. At deployment time, grounding is performed by an sLM by refactoring only the execution-specific components, enabling skills to be adapted to the target execution context with minimal modification. To enable systematic skill reuse across diverse robots, RECENT adopts a skill ontology that explicitly structures each skill according to shared semantic definitions (Tenorth & Beetz, 2017). The ontology encodes

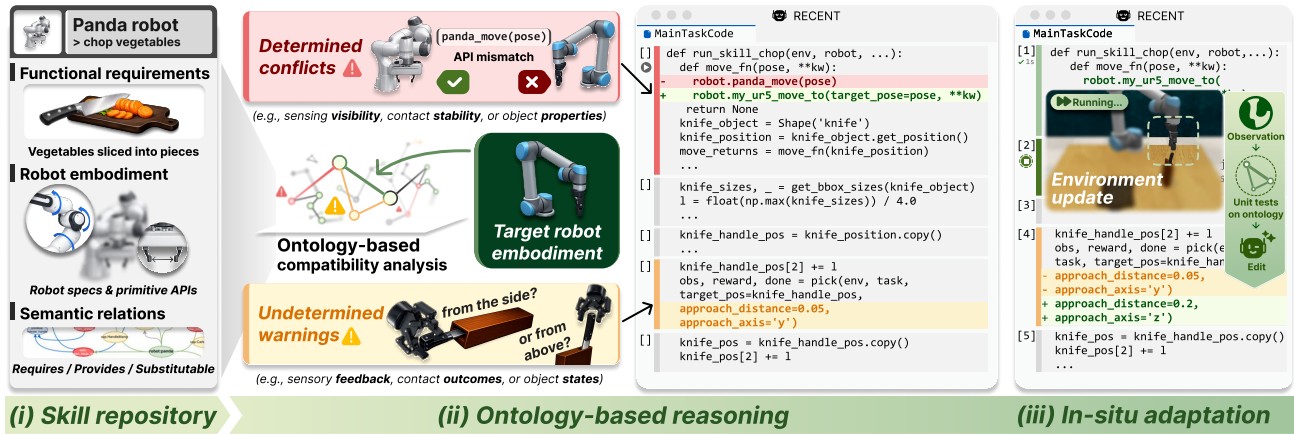

*Figure 2.* Overview of RECENT. (i) *Offline skill repository* stores reusable skill code with ontology-based metadata specifying functional intent, robot embodiment, and semantic relations. (ii) *Ontology-based reasoning* maps each skill to the target robot, diagnosing determined conflicts and undetermined warnings. (iii) *In-situ adaptation* patches code at execution time using environment feedback.

a skill's functional requirements (e.g., its definition, pre-conditions, and effects), robot embodiment profiles (e.g., available primitive APIs), and semantic relations between them (e.g., requires, provides, and substitutable-by). Based on this representation, we maintain an offline repository of reusable skills, where each skill is generated as executable code using cloud-scale LLMs and annotated with ontology-based metadata. All skills in the repository are validated on a commonly used robot, such as *Franka Emika Panda*, to ensure functional correctness prior to deployment. By construction, this representation enables skills to be systematically grounded across diverse deployment conditions.

Skill grounding in RECENT is performed at deployment time based on the execution context, which we decompose into determined and undetermined factors depending on their resolvability before execution. Before execution, embodiment-related factors are largely determined by the known capabilities of the target robot. Accordingly, RECENT leverages the skill ontology to either select directly compatible skills or identify substitutable code snippets to resolve embodiment mismatches. In these cases, an sLM performs localized code editing, such as replacing API calls, adjusting parameters, or rewiring interfaces, while preserving the overall functional structure for efficient reuse. During execution, environment-related factors, such as object properties, contact dynamics, partial observability, and scene constraints, may remain undetermined. To handle these factors, RECENT monitors execution-time environmental feedback and performs anticipatory in-situ code patching on yet-to-be-executed code fragments when a skill is predicted to violate its expected behavior. In this process, lightweight autonomous validation checks are inserted for undetermined factors to guide the sLM in deciding when and where patching is required, enabling uninterrupted

long-horizon execution without restarting the entire skill.

Figure 2 provides an overview of RECENT, which operates in three phases. Prior to deployment, we construct (i) an *offline skill repository*, where reusable skill code is generated using an LLM based on a skill ontology and validated for executability on a general-purpose robot. At deployment time, RECENT performs skill grounding through two additional phases: (ii) *Ontology-based reasoning* for embodiment gaps, which enables an sLM to perform targeted pre-execution editing for determined, embodiment-specific factors; and (iii) *In-situ adaptation* to environmental variations, which enables the sLM to perform on-demand patching for undetermined, environment-dependent factors.

### 4.1. Offline skill repository

The offline skill repository serves as the foundation for systematic skill reuse in RECENT. We adopt a skill ontology as a logical knowledge base that encodes skill semantics, robot embodiment capabilities, and semantic relations between them following (Tenorth & Beetz, 2017). Specifically, the ontology captures skill's functional intent, including preconditions, and expected effects, as well as robot embodiment profiles, such as available primitive APIs and execution constraints. In addition, it encodes semantic relations between skills and capabilities, including requires, provides, and substitutable-by. Each skill in the repository is generated as executable code using a cloud-scale LLM and is structured to separate semantic intent from execution-specific components. All skills are validated offline on a representative embodiment to ensure functional correctness and basic executability.

**Skill ontology querying.** Based on the skill ontology, we define an ontology query operator $\mathcal{Q}$ that extracts a robot-conditioned skill specification by retrieving ontology facts consistent with both the skill definition and the target embodiment. For a skill $\chi \in \mathcal{X}$ and a robot $r \in \mathcal{R}$, we define:

$$\mathcal{Q}(\chi, r) = \langle \mathcal{C}(\chi), \mathcal{P}(\chi), \mathcal{E}(\chi), \mathcal{I}(\chi, r) \rangle. \tag{2}$$

Here, $\mathcal{C}(\chi)$ denotes the set of required capabilities and operational constraints specified for skill $\chi$ (e.g., gripper type, sensing modalities), $\mathcal{P}(\chi)$ denotes preconditions that must hold before execution (e.g., object visibility, reachable poses, collision-free workspaces), and $\mathcal{E}(\chi)$ denotes the expected effects after execution (e.g., object pose changes, grasp success, state transitions). Crucially, $\mathcal{I}(\chi, r)$ denotes a robot-conditioned primitive API interface derived from the ontology that attempts to satisfy the required capabilities in $\mathcal{C}(\chi)$ under the capability profile of robot $r$. If no interface fully satisfies $\mathcal{C}(\chi)$, the remaining unmet requirements are exposed as embodiment gaps and deferred to the diagnosis operator in *Ontology-based reasoning* (Section 4.2). The resulting interface specifies typed inputs, outputs, and control parameters, and serves as an explicit grounding contract between abstract skill intent and embodiment-specific execution. This ontology-query formulation provides a specification that enables LLMs to generate skill code enriched with structured metadata.

**LLM-based skill generation and validation.** Let $r^{\text{src}}$ denote a general-purpose robot embodiment (e.g., *Franka Emika Panda*), which serves as a reference robot for offline skill validation. For each skill $\chi$, we query $\mathcal{Q}(\chi, r^{\text{src}})$ and use a cloud-scale LLM $\pi_{\text{LLM}}$ to generate an executable skill implementation $\tilde{\chi}$ enriched with ontology-aligned metadata. The resulting offline skill repository $\tilde{\mathcal{X}}$ is defined as the set of skills that satisfy canonical validation:

$$\tilde{\mathcal{X}} = \{\tilde{\chi} \mid \chi \in \mathcal{X}, \tilde{\chi} = \pi_{\text{LLM}}(\mathcal{Q}(\chi, r^{\text{src}})), \\ \text{Validate}(\tilde{\chi}, r^{\text{src}}) = 1\}. \tag{3}$$

The validation procedure $\text{Validate}(\cdot)$ executes $\tilde{\chi}$ on $r^{\text{src}}$ and verifies consistency with its specification by checking satisfaction of testable preconditions in $\mathcal{P}(\chi)$ and alignment between observed outcomes and the expected effects in $\mathcal{E}(\chi)$. Only validated skills are stored as reusable entries in the offline skill repository for deployment-time grounding.

## 4.2. Ontology-based reasoning for embodiment gaps

At deployment time, RECENT grounds skills to a target robot by reasoning over embodiment mismatches through the ontology. We categorize deployment-time uncertainties into determined embodiment factors and undetermined environmental factors. This phase resolves determined factors before execution via localized code editing, while deferring undetermined factors to *in-situ adaptation* (Section 4.3).

**Ontology-based compatibility analysis.** Given a target robot $r^{\text{tgt}}$, RECENT queries the skill ontology to assess the compatibility of each retrieved skill with the target embodiment. Specifically, for a candidate skill implementation $\tilde{\chi}$, we evaluate whether the capability profile of $r^{\text{tgt}}$ satisfies the required capabilities $\mathcal{C}(\chi)$ and whether the robot's available primitive APIs are consistent with the skill's interface on the ontology. If all required capabilities are directly supported by $r^{\text{tgt}}$, the skill is deemed compatible and selected without modification. Otherwise, RECENT performs ontology-based embodiment gap analysis by decomposing unmet capability requirements into determined and undetermined factors. Given $\tilde{\chi}$ and $r^{\text{tgt}}$, we define an ontology-based diagnosis operator $\mathcal{D}$ as:

$$\mathcal{D}(\tilde{\chi}, r^{\text{tgt}}) = \langle \Delta_{\text{det}}(\tilde{\chi}, r^{\text{tgt}}), \Delta_{\text{und}}(\tilde{\chi}, r^{\text{tgt}}) \rangle. \tag{4}$$

Here, $\Delta_{\text{det}}$ denotes determined conflicts whose resolution is fully specified by ontology-derived substitutions, allowing corresponding code interfaces and API calls to be refactored prior to execution. In contrast, $\Delta_{\text{und}}$ denotes undetermined, environment-contingent warnings (e.g., sensing visibility, contact stability, object physical properties) whose satisfaction cannot be certified prior to execution and must be validated online during *in-situ adaptation* (Section 4.3).

**Localized editing for determined conflicts.** For determined conflicts $\Delta_{\text{det}}(\tilde{\chi}, r^{\text{tgt}})$, RECENT resolves grounding via localized code editing using sLM-based FIM (Bavarian et al., 2022). Each conflict specifies an editing target defined by the start and end locations of the code span to be modified, together with ontology-derived substitutions as hints. Accordingly, determined conflicts are represented as a set of infilling tasks:

$$\Delta_{\text{det}}(\tilde{\chi}, r^{\text{tgt}}) = \{(m_i, \kappa_i)\}_{i=1}^{N}, \tag{5}$$

where each infilling task $(m_i, \kappa_i)$ consists of a code snippet $m_i$ to be refactored and a set of ontology-derived substitution hints $\kappa_i$. Each $\delta_i$ is resolved locally by an sLM $\pi_{\text{sLM}}$ using FIM to generate a replacement snippet:

$$m_i' \leftarrow \pi_{\text{sLM}}\big(\psi^{\text{pre}}(m_i; \tilde{\chi}), \psi^{\text{suf}}(m_i; \tilde{\chi}), \kappa_i\big). \tag{6}$$

where $\psi^{\text{pre}}(m_i; \tilde{\chi})$ and $\psi^{\text{suf}}(m_i; \tilde{\chi})$ denote the prefix and suffix code segments surrounding $m_i$, respectively. The updated $\tilde{\chi}$ is obtained as $\tilde{\chi} \leftarrow [\psi^{\text{pre}}(m_i; \tilde{\chi}) \oplus m_i' \oplus \psi^{\text{suf}}(m_i; \tilde{\chi})]$, where $\oplus$ denotes concatenation.

## 4.3. In-situ adaptation to environmental variations

This phase addresses undetermined factors $\Delta_{\text{und}}(\tilde{\chi}, r^{\text{tgt}})$ whose validity cannot be resolved before execution and instead depends on execution-time environmental conditions. In contrast to determined conflicts, these factors are evaluated and resolved in-situ during skill execution.

**Unit-test for undetermined warnings.** For each undetermined warning in $\Delta_{\text{und}}(\tilde{\chi}, r^{\text{tgt}})$, RECENT instantiates a lightweight, localized validation check in the form of an autonomous unit test. Each unit test encodes expected preconditions or effects associated with a specific subsequent code segment and is continuously evaluated using execution-time observations (e.g., object states, contact outcomes, or sensory feedback). Importantly, these validation checks are applied only to code segments that have not yet been executed, enabling anticipatory detection of potential violations without interrupting ongoing execution. When a validation check is violated, the corresponding undetermined factor is determined and treated as an immediate patching target. If a unit test for $u \in \Delta_{\text{und}}(\tilde{\chi}, r^{\text{tgt}})$ is violated, we convert it into an observation-conditioned infilling task:

$$u \xrightarrow{o_t} (m, o_t). \tag{7}$$

where $m$ denotes the target code snippet to be refactored, and $o_t$ denotes the environment observation at timestep $t$.

**Execution-time adaptive patching.** Given a patching target $(m, o_t)$, RECENT performs localized execution-time adaptation via FIM. Formally, given the current skill code $\tilde{\chi}$, we generate a replacement snippet using $\pi_{\text{sLM}}$:

$$m' \leftarrow \pi_{\text{sLM}}\big(\psi^{\text{pre}}(m; \tilde{\chi}), \psi^{\text{suf}}(m; \tilde{\chi}), o_t\big), \tag{8}$$

and update the skill code by patching only the targeted snippet as $\tilde{\chi} \leftarrow [\psi^{\text{pre}}(m; \tilde{\chi}) \oplus m' \oplus \psi^{\text{suf}}(m; \tilde{\chi})]$. This formulation modifies only the invalidated snippet while preserving the remaining code structure and ongoing execution context.

## 5. Evaluation

### 5.1. Experimental setting

To evaluate the efficiency and robustness of skill grounding under deployment constraints, we assess RECENT on long-horizon embodied manipulation tasks where embodiment gaps and environmental variations cause incompatibilities between offline-generated skills and deployment-time execution conditions. These tasks are designed to examine whether RECENT, when deployed with an sLM, can ground reusable skills via localized code refactoring instead of policy regeneration, while sustaining continuous execution over long horizons.

**Environments.** We design four evaluation settings spanning two types of embodiment mismatches: (1) *Kinematic variation*, where the source and target robots differ in manipulator structure, such as link configurations, joint layouts, and degrees of freedom; and (2) *End-effector variation*, where the grasping mechanism differs, such as parallel-jaw versus vacuum grippers. For kinematic variation, we use

*Figure 3.* Evaluation settings. (*left*) *Kinematic variations*, transferring from the source Panda robot to the target robots UR5 and Sawyer. (*right*) *End-effector variations*, transferring from the Franka Hand to the Robotiq 2F-85 and vacuum grippers.

Franka Emika Panda as the source embodiment and evaluate deployment on UR5 and Sawyer manipulators, which differ in kinematic structure and joint configuration. All robots are implemented in CoppeliaSim (Rohmer et al., 2013), enabling consistent evaluation across heterogeneous kinematic and control interfaces. For end-effector variation, we use Panda equipped with a parallel-jaw gripper as the source, and evaluate deployment using a Robotiq 2F-85 parallel gripper and a vacuum gripper. These settings are implemented in Genesis (Genesis Authors, 2024), which supports accurate modeling of contact dynamics and suction-based manipulation. Figure 3 summarizes the embodiment settings for evaluation. Beyond embodiment mismatches, all scenarios incorporate environmental variations that introduce dynamic and partial observability, including object pose perturbations, interaction uncertainties, and sensing noise. This setup ensures that skill execution requires grounding under both embodiment gaps and environmental variations.

**Tasks.** We construct each task as a continual sequence of subtasks, forming long-horizon manipulation problems that require up to 54 sequential API calls to complete under an optimal execution path. Each subtask is adapted from existing manipulation benchmarks, including RLBench, VIMA, and RoboGen (James et al., 2020; Jiang et al., 2022; Wang et al., 2023c). (1) *Kinematic variation* consists of 20 tasks, each requiring an average of 24.6 API calls. Each task combines up to 5 subtasks selected from 4 subtask categories: single-object manipulation, inter-object manipulation, precision interaction, and tool use. (2) *End-effector variation* consists of 12 tasks, each requiring an average of 29 API calls. Each task consists of subtasks that require stable grasping and accurate interactions between objects and the gripper, including object placement, articulated object interaction, scene rearrangement, and tool use.

**Baselines.** For comparison, we categorize and evaluate baselines into three groups for a comprehensive evaluation. (1) Code-as-Policies: We include methods following the Code-as-Policies paradigm, which generate executable policy code from task descriptions. We evaluate Code as Poli-

*Table 1.* Performance on long-horizon manipulation tasks under *Kinematic* and *End-effector* variation settings. Higher values indicate better performance for SR and GC, whereas lower values indicate better performance for GO, EI, and IT. All results are averaged over five runs with different random seeds.

| METHOD | Kinematic Variation | | | | | End-effector Variation | | | | |
|---|---|---|---|---|---|---|---|---|---|---|
| | SR (% ↑) | GC (% ↑) | GO (% ↓) | EI (count ↓) | IT (sec ↓) | SR (% ↑) | GC (% ↑) | GO (% ↓) | EI (count ↓) | IT (sec ↓) |
| CAP | $17.00_{\pm2.09}$ | $41.11_{\pm2.49}$ | $102.01_{\pm2.86}$ | $4.72_{\pm0.13}$ | $25.65_{\pm1.33}$ | $21.97_{\pm2.54}$ | $36.33_{\pm7.93}$ | $79.31_{\pm2.45}$ | $3.18_{\pm0.38}$ | $55.62_{\pm0.85}$ |
| CAP-CODEV-R1 | $20.00_{\pm10.00}$ | $36.10_{\pm13.72}$ | $109.38_{\pm33.84}$ | $4.13_{\pm2.49}$ | $17.50_{\pm6.65}$ | $17.88_{\pm6.63}$ | $41.03_{\pm10.81}$ | $99.16_{\pm9.47}$ | $2.77_{\pm0.84}$ | $43.77_{\pm2.43}$ |
| SCoT | $14.50_{\pm6.47}$ | $38.38_{\pm2.31}$ | $107.08_{\pm28.69}$ | $4.40_{\pm1.82}$ | $28.86_{\pm10.81}$ | $15.83_{\pm1.86}$ | $54.19_{\pm1.86}$ | $85.35_{\pm12.50}$ | $1.80_{\pm0.22}$ | $41.87_{\pm9.31}$ |
| PROGPROMPT | $31.50_{\pm7.20}$ | $53.89_{\pm7.07}$ | $86.47_{\pm5.54}$ | $3.28_{\pm1.42}$ | $30.74_{\pm29.83}$ | $33.33_{\pm11.18}$ | $58.16_{\pm7.40}$ | $65.34_{\pm0.77}$ | $4.72_{\pm0.79}$ | $39.32_{\pm5.67}$ |
| ROBOINSPECTOR | $41.00_{\pm4.59}$ | $60.83_{\pm6.20}$ | $179.40_{\pm81.75}$ | $1.15_{\pm0.45}$ | $59.98_{\pm33.97}$ | $39.17_{\pm14.72}$ | $64.80_{\pm5.41}$ | $52.20_{\pm10.24}$ | $5.07_{\pm0.33}$ | $31.65_{\pm10.37}$ |
| REPAIRAGENT | $42.00_{\pm4.11}$ | $64.05_{\pm2.50}$ | $54.17_{\pm19.87}$ | $1.73_{\pm0.14}$ | $50.53_{\pm17.35}$ | $52.17_{\pm2.09}$ | $73.82_{\pm3.04}$ | $55.92_{\pm28.83}$ | $1.12_{\pm0.25}$ | $84.37_{\pm75.24}$ |
| AGENTLESS | $22.50_{\pm13.31}$ | $42.70_{\pm12.42}$ | $128.08_{\pm1.18}$ | $2.19_{\pm0.61}$ | $58.27_{\pm20.82}$ | $26.34_{\pm13.98}$ | $49.88_{\pm13.62}$ | $74.96_{\pm5.44}$ | $5.01_{\pm0.50}$ | $42.44_{\pm11.10}$ |
| **RECENT (OURS)** | $\mathbf{73.00_{\pm3.71}}$ | $\mathbf{81.76_{\pm1.07}}$ | $\mathbf{7.85_{\pm3.00}}$ | $\mathbf{0.72_{\pm0.35}}$ | $\mathbf{1.77_{\pm0.90}}$ | $\mathbf{82.50_{\pm1.86}}$ | $\mathbf{89.84_{\pm2.05}}$ | $\mathbf{2.51_{\pm0.08}}$ | $\mathbf{0.69_{\pm0.44}}$ | $\mathbf{2.34_{\pm0.93}}$ |
| CAP-CODEX | $67.50_{\pm8.48}$ | $80.45_{\pm4.81}$ | $84.29_{\pm25.78}$ | $1.23_{\pm0.51}$ | $7.99_{\pm4.11}$ | $74.85_{\pm2.97}$ | $87.81_{\pm1.74}$ | $41.69_{\pm8.28}$ | $0.60_{\pm0.08}$ | $10.29_{\pm4.33}$ |

cies (CaP) (Liang et al., 2023) and its variants: **CaP** (Liang et al., 2023) instantiated with an sLM, **CaP-CodeV-R1**, a distilled variant of CaP (Zhu et al., 2025), and **CaP-Codex**, which employs GPT-5.2-Codex (OpenAI, 2025). We additionally include **SCoT** (Li et al., 2025b), which augments CaP with explicit reasoning for policy code generation. (2) Embodied Agentic Programming: We include embodied agentic programming methods that synthesize and execute programs online and integrate policy code generation into the embodied perception-action feedback loop. We use **ProgPrompt** (Singh et al., 2022) and **RoboInspector** (Ying et al., 2025) as representative baselines. (3) Automated Program Repair: We include automated program repair methods that iteratively revise programs based on reactive, error-driven execution feedback. We use **RepairAgent** (Bouzenia et al., 2025) and **Agentless** (Xia et al., 2025) as representative baselines.

**Metrics.** We evaluate performance using 5 metrics aligned with the objective in Eq. (1), covering task success, grounding efficiency, and execution continuity. For robustness, we report (1) **Task Success Rate (SR)** (%), which measures whether an agent successfully completes all required subtasks of a given task, and (2) **Goal-Conditioned Success Rate (GC)** (%), which evaluates whether individual goal conditions are satisfied. For grounding efficiency, we measure (3) **Grounding Overhead (GO)** (%), defined as the ratio of the number of tokens generated by the sLM during skill grounding to the token length of the original skill code. For execution continuity, we report (4) the **Execution Interruption Count (EI)**, defined as the number of pauses in skill execution caused by failures or environmental mismatches at runtime, and (5) the **Idle Time (IT)**, measured as the cumulative duration required to resolve such interruptions before skill execution resumes.

**Implementation.** All experiments are implemented in Python 3.9. We use Qwen2.5-Coder-7B as the default sLM, accessed via HuggingFace (Wolf et al., 2019). All baseline methods are evaluated under identical configurations

and executed on off-the-shelf NVIDIA RTX 4090 GPUs. Additional experimental details are in Appendix C.

### 5.2. Main result

Table 1 presents a comparison of RECENT and eight competitive baselines on long-horizon embodied manipulation tasks across two skill grounding scenarios, *Kinematic variation* and *End-effector variation*, both evaluated in dynamic environments. Across both scenarios, all baselines are provided with the same set of reference skills, which are used for grounding and reuse to synthesize executable policy code for task execution. Under the same sLM setting, RECENT consistently outperforms all baselines in task success, grounding efficiency, and execution continuity. In terms of task performance, RECENT achieves the highest SR and GC, with 73.00% SR and 81.76% GC in *Kinematic variation*, and 82.50% SR and 89.84% GC in *End-effector variation*. For grounding efficiency, it attains the lowest GO of 7.85% and 2.51% in *Kinematic variation* and *End-effector variation*, respectively. RECENT further preserves execution continuity, maintaining near-zero EI of 0.72 and 0.69 and achieving the lowest IT of 1.77 sec and 2.34 sec across the two scenarios.

Averaged across *Kinematic variation* and *End-effector variation*, RECENT outperforms the strongest baseline RepairAgent, achieving higher task success with improvements of 30.67 pp in SR and 16.87 pp in GC, while reducing GO, EI, and IT by 49.87 pp, 0.72, and 65.40 sec respectively. Compared to the distilled sLM-based baseline **CaP-CodeV-R1**, RECENT significantly improves SR and GC by 58.81 pp and 47.24 pp, and reduces GO by 99.09 pp, EI by 2.75, and IT by 28.58 sec. At the same time, RECENT achieves performance comparable to the fully accessible LLM-based baseline **CaP-Codex**, with absolute differences of only 6.58 pp in SR and 0.21 in EI, while achieving a substantial 57.81 pp improvement in GO and a 7.09 sec reduction in IT.

Specifically, the *Kinematic variation* setting is designed to evaluate skill grounding under kinematic embodiment mis-

matches between source and target manipulators with different kinematic structures, including variations in joint configurations, degrees of freedom, and workspace geometry (e.g., Panda → UR5, Sawyer). Under this setting, embodiment mismatches primarily arise at the motion level, such as discrepancies in joint limits, feasible trajectory waypoints, and inverse kinematics solutions. This setting evaluates whether a skill can be grounded to respect target-specific kinematic constraints while preserving its functional semantics. The SCoT baseline relies on structured intermediate representations to guide code synthesis, but resolving motion-level kinematic mismatches in this formulation still requires the sLM to synthesize kinematically valid code without localized modification targets, whereas RECENT identifies deterministic kinematic conflicts and resolves them through minimal, ontology-guided refactoring. Compared to SCoT, RECENT improves SR by 58.50 pp and GC by 43.38 pp, while reducing GO from 107.08% to 7.85%.

In the *End-effector variation* setting, skill grounding is designed to evaluate adaptation under changes in the grasping mechanism. Embodiment mismatches arise from differences in end-effector interfaces, primarily involving substitutions of grasp and release primitives (e.g., parallel-jaw grasp versus suction attach and detach) and the corresponding low-level control APIs. This setting assesses whether end-effector-related adaptations can be localized while preserving functional semantics. Failures in this setting are often induced by contact outcomes and grasp uncertainty that emerge only at execution time, and the RoboInspector baseline repairs such failures through full code regeneration based on categorized error feedback, which results in excessive token overhead, whereas RECENT restricts modifications to ontology-identified end-effector-related code spans. RECENT improves SR and GC by 43.33 pp and 25.04 pp, while reducing GO by 49.69 pp, EI by 4.38, and IT by 29.31 sec compared to the RoboInspector baseline.

## 5.3. Ablation study

*Table 2.* Ablation study on sLM family and model scale

| FAMILY | MODEL SIZE | SR (% ↑) | GO (% ↓) | EI (count ↓) | IT (sec ↓) |
|---|---|---|---|---|---|
| QWEN2.5-CODER | 1.5B | 66.67$_{\pm0.00}$ | 4.81$_{\pm0.84}$ | 0.94$_{\pm0.03}$ | 7.83$_{\pm0.88}$ |
| | 3B | 83.33$_{\pm0.00}$ | 3.42$_{\pm0.21}$ | 0.50$_{\pm0.02}$ | 15.24$_{\pm1.75}$ |
| | 7B | 83.33$_{\pm0.00}$ | 2.64$_{\pm0.09}$ | 0.28$_{\pm0.01}$ | 2.62$_{\pm2.21}$ |
| CODEGEMMA | 2B | 66.67$_{\pm0.00}$ | 6.38$_{\pm0.47}$ | 1.04$_{\pm0.21}$ | 1.21$_{\pm0.52}$ |
| | 7B | 83.33$_{\pm0.00}$ | 3.22$_{\pm1.04}$ | 0.39$_{\pm0.23}$ | 19.40$_{\pm16.51}$ |

Table 2 reports the performance of RECENT across different sLM families (i.e., Qwen2.5-Coder (Hui et al., 2024) and CodeGemma (Zhao et al., 2024)) and model sizes ranging from 1.5B to 7B. This ablation evaluates the robustness of our framework across varying sLM architectures and model capacity levels. Across model families, performance remains largely consistent in terms of SR, GO, and EI. When using 7B models, both sLM families achieve identical SR

of 83.33%, while GO differs by only 0.58 pp (2.64% for Qwen2.5-Coder and 3.22% for CodeGemma) and EI by 0.11 (0.28 and 0.39), indicating that our framework is compatible with recently advancing code generation sLMs. IT shows larger variation across families and scales (2.62 sec for Qwen2.5-Coder 7B and 19.40 sec for CodeGemma 7B), reflecting differences in per-call inference latency across sLM implementations rather than differences in grounding behavior. When scaling down to lighter models, robustness is largely preserved at the 3B scale, whereas noticeable performance degradation emerges below 2B. Despite this degradation, even the smallest models consistently outperform baselines using 7B models, as shown in Table 1. These results are enabled by RECENT's decomposed grounding process, which offloads deployment-time mismatch diagnosis from the sLM to the offline-constructed ontology, whose compatibility relations deterministically identify conflicts and localize refactoring targets. By reducing the reasoning burden on the sLM, this design decouples skill grounding from model-specific reasoning capacity, enabling consistent performance across model families and scales.

*Table 3.* Ablation study on offline skill validation

| METHODS | SR (% ↑) | GO (% ↓) | EI (count ↓) | IT (sec ↓) |
|---|---|---|---|---|
| W/O. OFFLINE VALIDATION | 62.50$_{\pm3.54}$ | 14.95$_{\pm1.50}$ | 1.15$_{\pm0.10}$ | 1.62$_{\pm1.45}$ |
| **RECENT** | **72.50**$_{\pm3.54}$ | **8.49**$_{\pm8.04}$ | **0.95**$_{\pm0.00}$ | **1.67**$_{\pm2.23}$ |

**Ablation study on offline skill validation.** Table 3 shows the effect of offline skill validation in RECENT. We ablate offline skill validation by replacing the repository with skills that have undergone grounding once but are not canonically validated. Under this setting, performance degrades, with SR decreasing by 10.00 pp, GO increasing by 6.46 pp, and EI increasing by 0.20, while IT remains comparable. These results arise because skills without offline validation do not guarantee the functional correctness of their semantic intent, which can manifest as deployment-time mismatches and require the sLM to perform refactoring beyond the localized scope identified by the ontology. Offline validation reduces deployment-time grounding cost and supports more stable long-horizon execution. Even so, RECENT without offline skill validation remains competitive with the baselines reported in Table 1, reflecting the robustness of its deployment-time grounding procedure.

*Table 4.* Ablation study on RECENT

| METHODS | SR (% ↑) | GO (% ↓) | EI (count ↓) | IT (sec ↓) |
|---|---|---|---|---|
| W/O. ONTOLOGY GUIDANCE | 21.67$_{\pm7.45}$ | 2.35$_{\pm1.55}$ | 2.57$_{\pm0.59}$ | 4.14$_{\pm2.15}$ |
| → W/ API NAME SIMILARITY | 58.33$_{\pm0.00}$ | 3.78$_{\pm2.64}$ | 1.71$_{\pm0.48}$ | 1.14$_{\pm0.51}$ |
| W/O. IN-SITU ADAPTATION | 66.67$_{\pm0.00}$ | 1.77$_{\pm1.02}$ | 0.76$_{\pm0.00}$ | 5.91$_{\pm3.59}$ |
| → FULL RE-GENERATION | 26.67$_{\pm9.13}$ | 89.63$_{\pm1.40}$ | 2.44$_{\pm0.44}$ | 91.21$_{\pm2.50}$ |
| **RECENT** | **83.33**$_{\pm0.00}$ | **2.79**$_{\pm0.21}$ | **0.36**$_{\pm0.11}$ | **1.90**$_{\pm1.84}$ |

**Ablation study on RECENT components.** Table 4 reports the contribution of individual components of RECENT to overall performance. The **w/o. Ontology guidance** setting disables ontology-based guidance in phase (ii), forcing determined conflicts to be resolved through full code regeneration rather than FIM-based localized editing with ontology-derived substitution hints. Without ontology guidance, SR drops by 61.66 pp, showing that embodiment conflicts must be explicitly diagnosed and localized for effective grounding. The → **w/ API name similarity** setting restores localized FIM editing, but replaces ontology-derived substitutions with heuristic API-name matching. Although API-name similarity partially recovers performance, it still underperforms RECENT by 25.00 pp in SR, indicating that localized editing alone is insufficient when substitutions are not semantically grounded. This limitation is particularly evident for capability-level equivalences such as `open_gripper` → `deactivate_vacuum`. In the **w/o. In-situ adaptation** setting, anticipatory patching in phase (iii) is disabled, so environment-dependent mismatches are no longer resolved by patching yet-to-be-executed code spans during execution and are instead handled reactively after failures occur. Disabling in-situ adaptation decreases SR by 16.66 pp and increases EI by 0.40, showing that reactive recovery interrupts execution more often and reduces task completion reliability. The → **Full re-generation** setting replaces FIM-based localized editing in phases (ii) and (iii) with full code regeneration, while retaining ontology-guided diagnosis and validation. Without infilling, mismatches are resolved by regenerating the entire skill code rather than only the affected spans, increasing GO by 86.84 pp and IT by 89.31 sec while decreasing SR by 56.66 pp.

## 6. Conclusion

In this work, we present RECENT, a refactoring-centric agent framework that enables efficient skill grounding with sLMs under deployment constraints. By representing skills as executable code that explicitly separates invariant semantic intent from embodiment- and environment-specific execution bindings, RECENT enables code refactoring rather than full code regeneration during deployment. Extensive experiments demonstrate that RECENT significantly outperforms distilled sLM-based CaP baselines in success rate, grounding efficiency, and execution stability, while achieving task performance comparable to LLM-based CaP.

**Limitation and future work.** RECENT is scoped to deployment settings in which a shared and stable skill ontology is established through one-time offline construction and validation, and performs deployment-time grounding within the resulting ontology. Within this scope, our framework supports execution-binding edits and structural adaptations of skill logic, as long as the required behavior remains expressible using primitives and capability relations already represented in the ontology. Cases requiring ontology-level capability extension arise when the required capabilities are not represented in the ontology. Rather than deployment-time grounding, these cases correspond to incremental skill learning (Lee et al., 2024), where new capabilities must be acquired and integrated into the skill ontology. Extending RECENT to support such ontology-level expansion and persistent repository evolution remains an important future direction for enabling lifelong skill learning across evolving embodiments and deployment environments.

## Acknowledgement

This work was supported by Institute of Information & communications Technology Planning & Evaluation (IITP) grant funded by the Korea government (MSIT), (RS-2022-II220043, Adaptive Personality for Intelligent Agents, RS-2022-II221045, Self-directed multi-modal Intelligence for solving unknown, open domain problems, RS-2025-02218768, Accelerated Insight Reasoning via Continual Learning, RS-2025-25442569, AI Star Fellowship Support Program (Sungkyunkwan Univ.), RS-2026-25543726, Development of Leading Talent in Medical Domain-Specific Generative AI, RS-2026-25528384, Resource-Intensive AI Technologies Based on Sustainable GPU Integrated Platforms, RS-2019-II190421, Artificial Intelligence Graduate School Program (Sungkyunkwan University)), the National Research Foundation of Korea (NRF) grant funded by the Korea government (MSIT) (No. RS-2026-25474409), IITP-ITRC (Information Technology Research Center) grant funded by the Korea government (MSIT) (IITP-2025-RS-2024-00437633, 10%), IITP-ICT Creative Consilience Program grant funded by the Korea government (MSIT) (IITP-2026-RS-2020-II201821, 10%), the AI Computing Infrastructure Enhancement (GPU Rental Support) User Support Program funded by the Ministry of Science and ICT (MSIT), Republic of Korea (No. RQT-25-120157), and by Samsung Electronics Co., Ltd.

## Impact Statement

This paper presents work whose goal is to advance the field of Machine Learning. There are many potential societal consequences of our work, none of which we feel must be specifically highlighted here.

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

# A. Implementation Details of the Approach

## A.1. Skill Ontology Structure

Figure 4 illustrates the schema of the skill ontology, which connects robots, capabilities, skills, and primitives used for deployment-time reasoning. Figure 5 shows a partial instance graph of the skill ontology, where querying detects that the suction embodiment lacks the specific gripper interface required by open_gripper and infers a valid substitution to the embodiment-compatible primitive deactivate_vacuum.

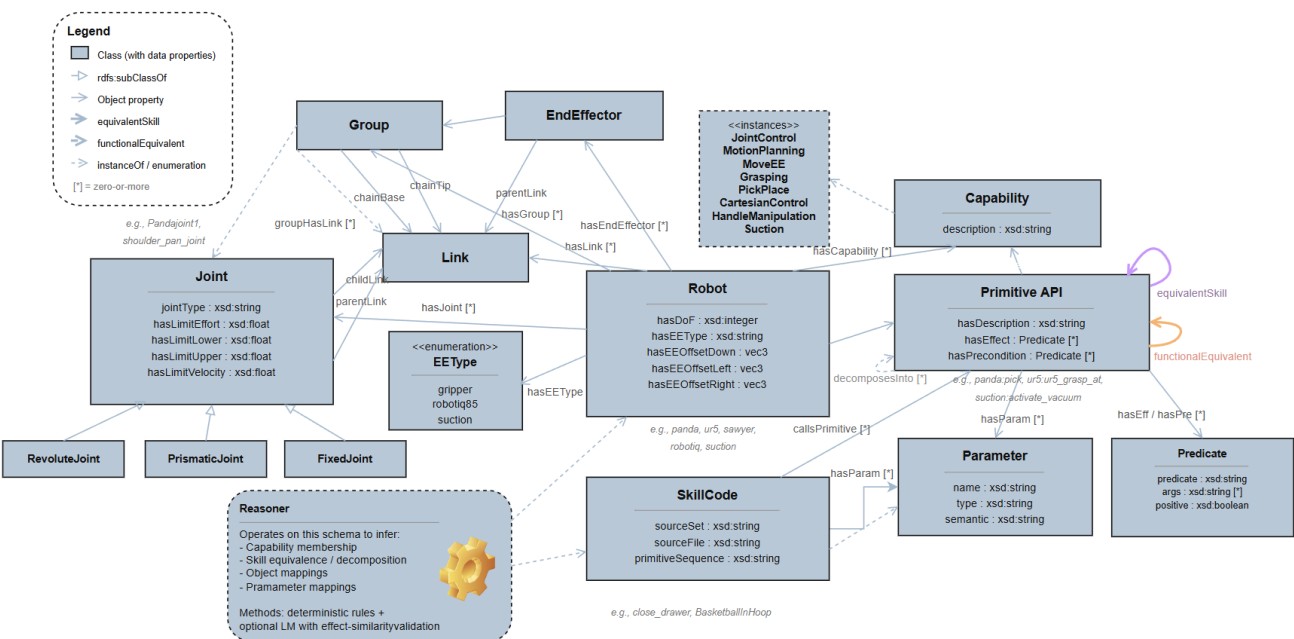

*Figure 4.* Schema of the skill ontology, defining classes and relations among skills, capabilities, robot embodiments, and primitives

## A.2. Skill Ontology Construction

**Pipeline inputs.** We construct the ontology from robot embodiment descriptions, primitive API specifications, and reference skill code using a deterministic pipeline. The ontology follows a shared schema adapted from prior robot knowledge representations (Tenorth & Beetz, 2017), and uses a fixed set of alignment rules to derive relations between robot capabilities, primitive APIs, and skill requirements. Each robot embodiment is annotated once with its embodiment profile and primitive API metadata, including available capabilities, parameter specifications, preconditions, and expected effects. Specifically, the pipeline integrates three sources of information. **Robot embodiment specifications** obtained from URDF and SRDF files provide robot structure information such as joints, links, planning groups, end-effectors, and kinematic constraints. **Primitive API specifications** defining executable robot interfaces are parsed to obtain callable interfaces, parameter signatures, capability requirements, preconditions, and expected effects. **Executable reference skill code** is analyzed to identify primitive call sites, parameter bindings, and control-flow structure. Among these, URDF and SRDF files are standard artifacts typically shipped with the robot, while primitive API metadata is provided once per embodiment as a lightweight, structured annotation.

**Deterministic relation instantiation.** All extracted information is represented as semantic triples within a shared ontology graph. Based on these triples, relations such as substitutableBy and functionalEquivalent are instantiated automatically through deterministic compatibility matching rules. For example, if two primitive APIs provide equivalent effects under compatible preconditions, the ontology instantiates a substitution relation between them. This construction avoids task-specific or robot-pair-specific substitution rules. Once the robot and primitive specifications are provided, the same pipeline instantiates the ontology relations automatically. The ontology additionally performs capability inference using deterministic rules defined over embodiment and API specifications. For example, robots equipped with valid planning

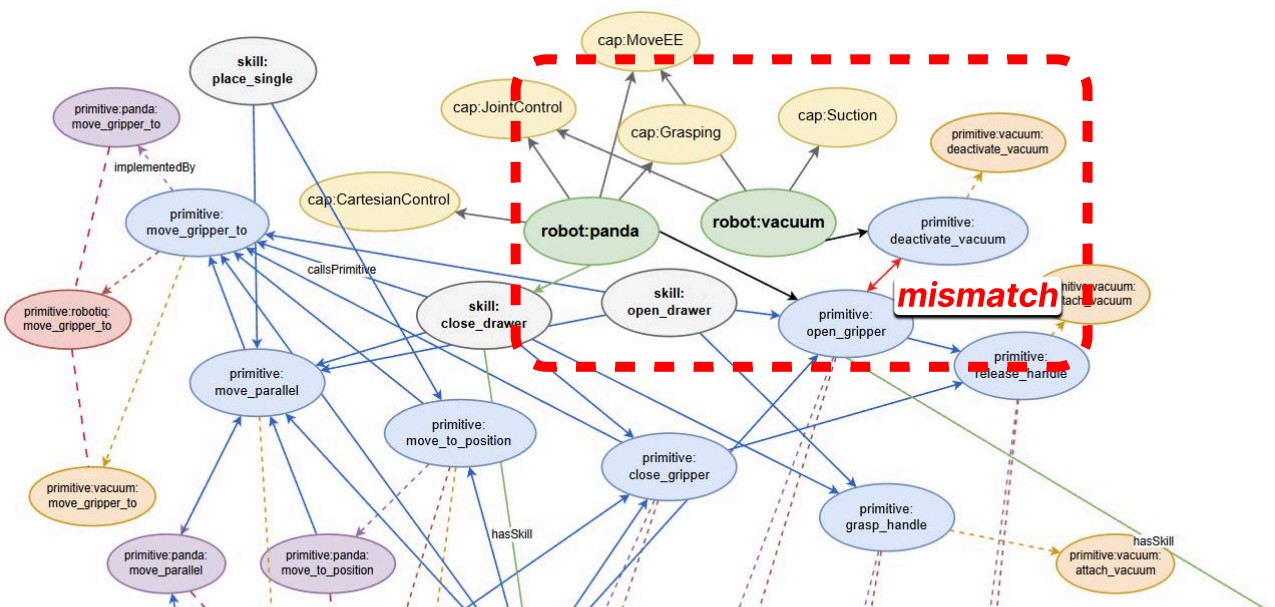

*Figure 5.* Partial instance graph of the skill ontology, illustrating ontology-based reasoning, where a capability mismatch is detected and resolved by substituting an embodiment-compatible primitive

groups and end-effectors are assigned motion-planning capabilities, while gripper-equipped embodiments are assigned grasp-related capabilities. Optionally, an sLM may provide auxiliary capability suggestions using primitive descriptions, but all inferred relations are validated against the existing ontology constraints before incorporation.

**One-time extension cost.** Once the robot embodiment and primitive specifications are provided, ontology construction is fully automated and implemented as a deterministic Python pipeline, requiring approximately 0.08 seconds across all experiments without task-specific manual edits. This confines the extension effort for a new embodiment to providing its URDF/SRDF files and primitive API metadata, without requiring task-specific or robot-pair-specific substitution rules. Moreover, because the ontology is shared across tasks, adding new skills does not require modifying the ontology itself, and newly generated skills can be directly validated and inserted into the offline skill repository.

### A.3. Skill Ontology Scale

**Ontology composition.** Table 5 summarizes the scale of the instantiated ontology used across all experiments. Each entry in the ontology is represented as a *(subject, predicate, object)* triple, encoding relations among robots, primitives, capabilities, parameters, preconditions, and effects. Although compact, the ontology covers multiple robot embodiments, primitive APIs, capability annotations, and primitive-level specifications, including parameters, preconditions, and expected effects. These relations provide the ontology-level evidence used to localize and guide deployment-time refactoring.

**Coverage of derived relations.** The 40 substitution and equivalence relations reported in Table 5 span both cross-embodiment primitive substitutions and within-embodiment functional equivalences over the 40 primitive APIs and 29 capabilities. Cross-embodiment relations connect primitives across grippers and manipulators with different interfaces, such as `close_gripper` $\leftrightarrow$ `activate_vacuum` for grasp acquisition and `open_gripper` $\leftrightarrow$ `deactivate_vacuum` for release. Together with the 1,232 semantic triples encoding robot-capability, primitive-precondition, and primitive-effect bindings, these relations provide sufficient coverage for the deployment-time grounding scenarios evaluated in our experiments. All relations are derived by the ontology construction pipeline described in Appendix A.2 rather than authored per task or robot pair.

*Table 5.* Scale of the instantiated ontology used in our experiments.

| Ontology Component | Count |
|---|---|
| Robot embodiments | 5 |
| Primitive APIs | 40 |
| Capabilities | 29 |
| Semantic triples | 1,232 |
| Parameter specifications | 82 |
| Preconditions | 67 |
| Expected effects | 66 |
| Substitution / equivalence relations | 40 |

## A.4. Refactoring Pipeline Details

We provide additional implementation details of the refactoring pipeline used in RECENT. The goal of this pipeline is not to regenerate an entire skill program, but to refactor only localized execution bindings that are inconsistent with the target embodiment or the current environment. At a high level, RECENT first detects a deployment-time mismatch through ontology-based compatibility analysis or execution-time unit tests, converts it into a structured infilling task augmented with ontology-derived substitution hints, and resolves it through FIM-based span replacement. The patched code is then validated and either executed or further adapted using runtime observations. This unified procedure is used for both pre-execution embodiment grounding (Section 4.2) and in-situ adaptation during execution (Section 4.3). The implementation, execution scripts, and configuration files used in our experiments are included in the supplementary material. Below, we describe the four stages of the pipeline: (1) detecting a mismatch and diagnosing it as a determined conflict or an undetermined warning, (2) constructing an intermediate refactoring unit from the detected mismatch, (3) converting it into an FIM prompt and generating a replacement span with the sLM, and (4) validating the patched code and adapting it further during execution.

**Mismatch detection and diagnosis.** Before any code refactoring, RECENT identifies where and why the current skill code is inconsistent with the deployment context. At pre-execution time, the ontology-based diagnosis operator $\mathcal{D}(\tilde{\chi}, r^{\mathrm{tgt}}) = \langle \Delta_{\mathrm{det}}, \Delta_{\mathrm{und}} \rangle$ (Eq. 4) compares the skill's required capabilities against the capability profile of $r^{\mathrm{tgt}}$, decomposing unmet capability requirements into determined conflicts $\Delta_{\mathrm{det}}$, which can be resolved by ontology-derived substitutions, and undetermined, environment-contingent warnings $\Delta_{\mathrm{und}}$, whose satisfaction depends on execution-time conditions. At execution time, each warning $u \in \Delta_{\mathrm{und}}$ is instantiated as an autonomous unit test that monitors the corresponding precondition or effect against incoming observations $o_t$. When such a unit test is violated, the corresponding undetermined factor is determined and converted into an observation-conditioned infilling task $u \xrightarrow{o_t} (m, o_t)$ (Eq. 7). Each detected mismatch, whether from pre-execution diagnosis or in-situ violation, is then passed to the next stage along with its ontology-derived substitution hints (for $\Delta_{\mathrm{det}}$) or its triggering observation (for $\Delta_{\mathrm{und}}$).

**Intermediate refactoring unit.** Given a mismatch detected in the previous stage, RECENT converts it into a structured intermediate refactoring unit. Each unit is extracted from the original skill code and consists of the target code span, its surrounding context, the span type, and grounding metadata:

$$z = \langle m, \psi_{\mathrm{pre}}, \psi_{\mathrm{suf}}, \rho, \kappa \rangle,$$

where $m$ denotes the editable target span, $\psi_{\mathrm{pre}}$ and $\psi_{\mathrm{suf}}$ denote the prefix and suffix surrounding the span, $\rho$ denotes the span type, and $\kappa$ denotes structured grounding metadata. The span type $\rho$ specifies whether the target corresponds to an API substitution site, an interface or parameter binding point, a preparatory step, or a warning-triggering statement. The metadata $\kappa$ includes the target embodiment, ontology-derived substitute APIs, interface constraints, parameter constraints, and, when applicable, the execution-time observation that triggered the patch. For determined conflicts, the span $m$ is typically an API call, argument binding, or wrapper interface whose substitute can be inferred from the ontology. For undetermined warnings, the span $m$ is selected from yet-to-be-executed statements whose expected preconditions or effects are violated by execution-time observations. This representation allows the sLM to operate on localized execution bindings while preserving the global control structure of the original skill.

**FIM prompt construction and span generation.** The refactoring unit $z$ is then converted into a localized FIM-based infilling task. The sLM receives $\psi_{\text{pre}}$ and $\psi_{\text{suf}}$ as the prefix and suffix of the FIM template, together with the structured grounding hints in $\kappa$ rendered as inline comments preceding the target span. These hints include the target embodiment, ontology-derived substitute APIs, interface and parameter constraints, and, for in-situ adaptation, the execution-time observation that triggered the patch. The span type $\rho$ further controls how the hint comments are organized, so that the sLM is presented with the minimal set of grounding cues relevant to the current edit. Given this prompt, the sLM generates a replacement span $m'$, which is constrained to fill the region between $\psi_{\text{pre}}$ and $\psi_{\text{suf}}$. This keeps generation length proportional to the size of the edit rather than the size of the skill, and prevents the sLM from rewriting unrelated code regions.

**In-situ validation and patching.** The generated span $m'$ is reinserted into the skill code at the position of $m$, producing a patched skill. In our implementation, the patched code is then re-evaluated against the unit tests that originally detected the violation (Section 4.3), using the latest execution-time observation to re-check the relevant precondition or effect. If the patched span passes this check, execution resumes from the same point without restarting prior steps, preserving the long-horizon execution context. If it fails, the corresponding undetermined factor is treated as an immediate patching target, and a new refactoring unit is constructed by repeating the previous two stages on a narrower or adjacent span. This loop terminates either when validation succeeds or when consecutive failed patches reach a preset limit, at which point the skill is reported as unrecoverable.

## B. Baseline Implementations

### B.1. CaP-variants

**Code-as-Policies.** We implement Code-as-Policies (CaP) as a minimal lower-bound baseline by directly introducing code generation into an embodied agent without any additional reasoning, planning, or execution-aware mechanisms. The agent is prompted with a task description, a list of available primitive skills, and a small number of in-context examples, and generates a complete executable policy as a single Python program. The generated code is executed directly in the simulator without intermediate validation or localization. Upon execution failure, simulator error messages are appended to the prompt and the entire program is regenerated, without performing partial edits or structured repair.

**CaP-CodeV-R1.** CaP-CodeV-R1 follows the same implementation as the CaP baseline while replacing the backbone language model with `zhuyaoyu/CodeV-R1-Distill-Qwen-7B`. The prompting format, in-context examples, and full-program regeneration strategy are kept identical to CaP, ensuring that the overall framework and interaction protocol remain unchanged. This baseline isolates the effect of a distilled code-oriented language model with limited capacity, without introducing any additional embodied reasoning or execution-aware mechanisms.

**SCoT.** We implement SCoT as a baseline that produces structured intermediate outputs prior to code generation. Instead of relying on external symbolic resources, the agent generates structured representations that specify primitive skill substitutions and parameter mappings before synthesizing an executable policy. The final code is generated based on this intermediate structure and executed directly in the simulator. This implementation follows the original SCoT formulation of generating structured guidance before code synthesis, without applying localized code modification or execution-aware refinement.

**CaP-Codex.** CaP-Codex follows the same implementation as the CaP baseline, while using `GPT-5.2-Codex` as the backbone language model with its default reasoning setting. The overall prompting structure and full-program generation protocol are largely aligned with CaP, but we provide richer and less-processed inputs that are feasible for a stronger model to interpret, including raw numerical state information (e.g., object positions and quaternions) and unfiltered simulator error logs. Unlike small language models, Codex can robustly handle these raw signals without inducing severe hallucination or performance degradation. This baseline evaluates the effect of a large-scale, code-specialized model within the CaP paradigm, without introducing additional planning, localization, or execution-aware refinement mechanisms.

### B.2. Embodied Code Agent

**ProgPrompt.** We adapt ProgPrompt as a one-shot code generation baseline by providing the source robot's reference code as an in-context example within the prompt. The agent is provided with a Pythonic program header that imports available actions and their expected parameters, along with a list of environment objects, enabling the agent to generate executable task plans as function implementations. The agent is instructed to translate the reference code from the source robot's

primitive skills to the target robot's primitive skills while preserving task logic. Generated code is executed directly in the simulator.

**RoboInspector.** We adopt RoboInspector's behavior detection taxonomy (Nonsense, Disorder, Infeasible, Badpose) as a pre-execution filter for generated code. The detector analyzes code structure, action sequences, and constraint patterns before simulation. When unreliable behaviors are detected or execution fails, the feedback refiner regenerates code with categorized error descriptions, enabling behavior-aware repair prompts.

### B.3. Automatic Program Repair Agents

**RepairAgent.** We adopt RepairAgent as a baseline framework by implementing its core components for embodied policy repair, while keeping state transitions and tool execution programmatic. In our implementation, the finite state machine deterministically switches between generate, repair, and revise based on simulator execution outcomes. A middleware executes the generated policy in the embodied environment and exposes structured signals as tool outputs. These outputs are incrementally appended to a dynamic prompt, together with the current scene and task constraints, enabling failure-localized edits. As a result, the repeated execution and repair state transition can become a bottleneck in long-horizon tasks under noisy simulator feedback.

**Agentless.** We implement Agentless with CaP by constructing constraint-aware prompts that include robot-specific primitive skill specifications, available skill primitives, scene object information, and workspace bounds. The prompt explicitly specifies physical constraints and available actions, allowing the agent to generate executable code without iterative refinement. Upon execution failure, error feedback is appended to the prompt and the code is regenerated at an appropriate granularity, targeting individual code lines, functions, or entire files depending on the localization result.

# C. Experimental Setting

We evaluate the agents under four evaluation settings spanning two types of embodiment mismatches: (1) *Kinematic variation* and (2) *End-effector variation*, all implemented in simulation environments. We further evaluate the agents in real-world environments, which are described in a separate subsection. Table 6 summarizes their key differences, and we describe each setting in detail in the following sections.

*Table 6.* Evaluation settings in simulation

| Variation Type | Key Property | Examples | # of Settings |
|---|---|---|---|
| *Kinematic variation* | Differences in manipulator structure | Link configuration, joint layout, degrees of freedom | 2 |
| *End-effector variation* | Differences in grasping mechanism | Parallel-jaw gripper, vacuum gripper | 2 |

## C.1. Kinematic variation

In the *Kinematic variation* setting, the agent is evaluated on its ability to adapt base skills to target environments with different kinematic structures, such as link configurations, joint layouts, or degrees of freedom. This requires the agent to abstract task-relevant motion patterns from embodiment-specific details and re-instantiate them under altered morphological constraints.

**Evaluation Scenario.** We instantiate two evaluation scenarios for the *Kinematic variation* setting, using the Franka Emika Panda (Panda) as the source embodiment and evaluating transfer to the UR5 and Sawyer manipulators, respectively. The example scene for the scenarios are depicted in Figure 6, with the summary provided in Table 7.

*Table 7.* Summary of kinematic variation scenarios

| Variation Type | Source | Target |
|---|---|---|
| *Kinematic variation* | Panda | UR5 |
| | Panda | Sawyer |

**Task Settings.** We construct each task as a continual stream of up to five subtasks sampled from a pool of four subtask groups. Each subtask group targets a distinct aspect of manipulation commonly evaluated in embodied intelligence benchmarks. By composing these subtasks into a stream, we generate long-horizon tasks that require diverse manipulation skills. The subtask groups are as follows:

- **Single-Object Manipulation**. The agent manipulates a single target object in isolation. Tasks in this category focus on fundamental motor skills such as grasping, lifting, and repositioning.

- **Inter-Object Manipulation**. The agent manipulates objects in relation to other objects in the scene. This requires understanding spatial relationships and coordinating actions that involve multiple entities, such as placing one object inside or on top of another.

- **Precision Interaction**. The agent performs tasks that demand high accuracy in end-effector positioning and orientation. Small deviations can lead to task failure, requiring fine-grained control over pose and motion trajectories.

- **Tool-use**. The agent leverages tools to accomplish goals that cannot be achieved through direct manipulation. The agent must recognize and effectively use the tool to accomplish the task.

Each subtask group consists of several individual subtasks, distinguished by their objectives and scene layouts. We implement the subtasks for the *Kinematic variation* setting using the CoppeliaSim engine (Rohmer et al., 2013). The individual subtasks for each group are listed in Table 8.

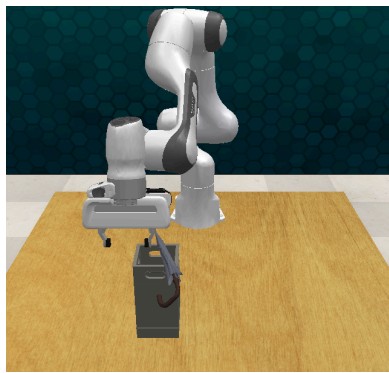

(a) Example of Panda (source)

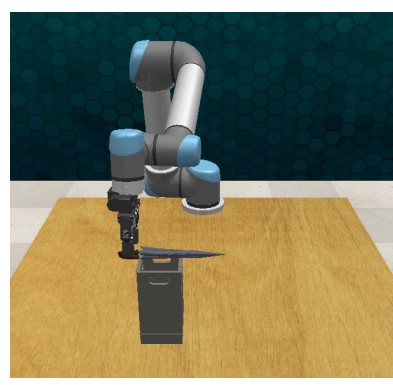

(b) Example of UR5 (target)

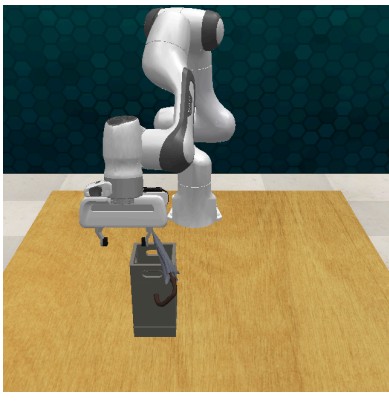

(c) Example of Panda (source)

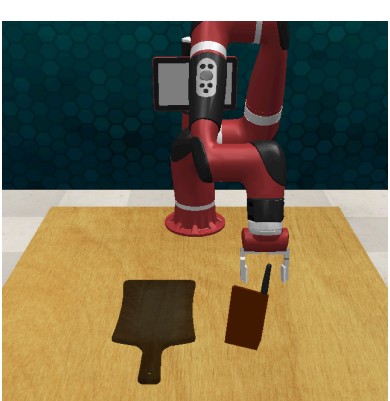

(d) Example of Sawyer (target)

*Figure 6.* Example scene of kinematic variation scenarios

*Table 8.* Description of kinematic variation subtasks

| Variation Type | Subtask Type | Subtask Instruction |
|---|---|---|
| *Kinematic variation* | Single-Object Manipulation | Turn off the lamp
Turn on the lamp
Pick and lift object
Push the button
Close the box
Turn on the TV
Change the channel |
| | Inter-Object Manipulation | Put basketball into hoop
Place phone on base
Put rubbish in bin
Take meat off grill
Place meat on grill
Put knife on chopping board |
| | Precision Interaction | Insert money into safe
Insert umbrella into stand
Unplug the charger |
| | Tool-use | Reach and drag object with stick
Wipe the desk
Hit ball with cue |

**Primitive APIs.** Table 9 summarizes the set of executable primitive APIs and their corresponding instruction templates designed for various robotic platforms within the RLBench environment. These APIs provide a structured interface for robot control at the function-level, such as picking, placing, and alignment.

*Table 9.* Executable APIs in RLBench

| | Kinematic Type | Template | Example |
|---|---|---|---|
| API | Panda | Pick [Object] | `pick(env, task, target_pos=[x,y,z], approach_axis='z')` |
| | | Place [Receptacle Object] | `place(env, task, target_pos=[x,y,z], approach_axis='z')` |
| | | Move [Object] | `move(env, task, target_pos=[x,y,z], timeout=FLOAT)` |
| | | Push [Object] | `push(env, task, target_pos=[x,y,z], approach_axis='z')` |
| | | Align To Quaternion [Object] | `align_to_quaternion(env, task, quaternion=q, ...)` |
| | | Align Two Axes | `align_two_axes(env, task, axis_dirs=(1,1), ...)` |
| | | Open Gripper | `open_gripper(env, task)` |
| | | Close Gripper | `close_gripper(env, task)` |
| | UR5 | Ur5 Grasp At [Object] | `ur5_grasp_at(env, task, grasp_pos=[x,y,z], timeout_s=5.0)` |
| | | Ur5 Release At [Receptacle Object] | `ur5_release_at(env, task, place_pos=[x,y,z], timeout_s=5.0)` |
| | | Ur5 Move To [Object] | `ur5_move_to(env, task, target_pos=[x,y,z], timeout_s=3.0)` |
| | | Ur5 Align Gripper [Object] | `ur5_align_gripper(env, task, reference_quat=q, ...)` |
| | | Open Ur5 EE | `open_ur5_ee(env, task, gripper_open=1.0, velocity=0.2)` |
| | | Close Ur5 EE | `close_ur5_ee(env, task, gripper_close=0.0, velocity=0.2)` |
| | Sawyer | Sawyer Pick [Object] | `sawyer_pick(env, task, target_object=obj, target_pos=[x,y,z])` |
| | | Sawyer Place [Receptacle Object] | `sawyer_place(env, task, place_pos=[x,y,z])` |
| | | Sawyer Move To [Object] | `sawyer_move_to(env, task, target_pos=[x,y,z])` |
| | | Sawyer Align Gripper [Object] | `sawyer_align_gripper(env, task, reference_quat=q, ...)` |
| | | Sawyer Open Gripper | `sawyer_open_gripper(env, task, amount=1.0, velocity=0.2)` |
| | | Sawyer Close Gripper | `sawyer_close_gripper(env, task, amount=0.0, velocity=0.2)` |

## C.2. End-effector variation

In *End-effector variation* setting, the agent is evaluated on its ability to adapt base skills to target environments with different grasping mechanisms, such as parallel-jaw grippers versus vacuum grippers. This requires the agent to generalize manipulation intent beyond specific grasp geometries and adjust grip strategies, and approach vectors accordingly.

**Evaluation Scenario.** We instantiate two evaluation scenarios for the *end-effector variation* setting, using the Franka Emika Panda with its default parallel-jaw gripper, Franka Hand, as the source embodiment. The first scenario evaluates transfer to a target environment with identical kinematics but equipped with a vacuum gripper; the second evaluates transfer to a Panda manipulator with a Robotiq 2F-85 gripper. The example scene for the scenarios are depicted in Figure 7, with the summary provided in Table 10.

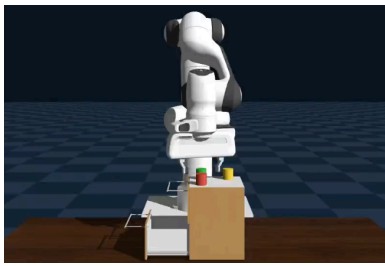 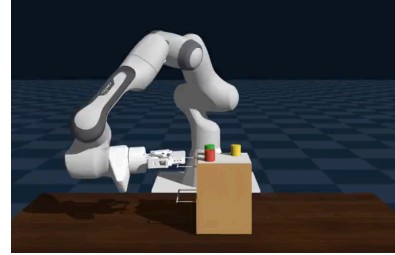

(a) Example of parallel-jaw gripper (source)    (b) Example of 2F-85 gripper (target)

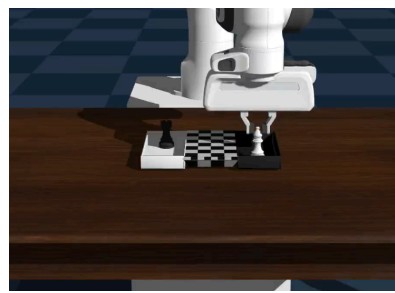 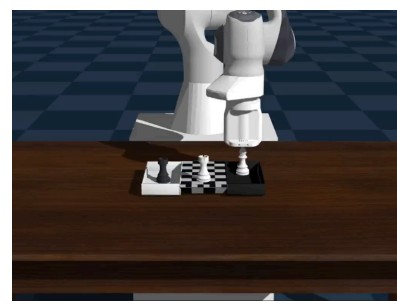

(c) Example of parallel-jaw gripper (source)  (d) Example of vacuum gripper (target)

*Figure 7.* Example scene of end-effector variation scenarios

*Table 10.* Summary of end-effector variation scenarios

| Variation Type | Source | Target |
|---|---|---|
| *End-effector variation* | Panda + Parallel-jaw gripper | Panda + Vacuum gripper |
| | Panda + Parallel-jaw gripper | Panda + 2F-85 gripper |

**Task Settings.** We construct each task as a continual stream of up to five subtasks sampled from a pool of four subtask groups, akin to the *Kinematic variation* setting. The specific subtask groups are as follows:

- **Object Placement**. The agent picks up a target object and places it at a specified location. This involves identifying the object, grasping it appropriately, and positioning it according to spatial instructions.

- **Articulated Object Interaction**. The agent interacts with objects that have movable joints, such as doors or drawers. The agent must understand the kinematic constraints and apply appropriate forces along the correct axis of motion.

- **Scene Rearrangement**. The agent rearranges multiple objects to match a target configuration. This involves planning a sequence of pick-and-place actions while considering dependencies and spatial relationships among objects.

- **Tool-use**. The agent leverages tools to accomplish goals that cannot be achieved through direct manipulation. The agent must recognize and effectively use the tool to accomplish the task.

Each subtask group consists of several individual subtasks, distinguished by their objectives and scene layouts. We implement the subtasks for the *End-effector variation* setting using the Genesis simulator (Genesis Authors, 2024). The individual subtasks for each group are listed in Table 11.

| Variation Type | Subtask Type | Task Instruction |
|---|---|---|
| *End-effector variation* | Object Placement | Place cubes into box
Place object into target tray
Place object into upside-down tray |
| | Articulated Object Interaction | Put fruits in hinge box
Place cylinders in prismatic drawer
Insert cylinders into drawers by color |
| | Scene Rearrangement | Stack cubes into tower
Arrange fruits without contact
Insert object into slot |
| | Tool-use | Sweep chess pieces into boxes
Sweep to sort chess pieces by color
Pour liquid into cup |

*Table 11.* Description of end-effector variation subtasks

**Primitive APIs.** Table 12 summarizes the set of executable primitive APIs and their corresponding instruction templates designed for various robotic platforms within the Genesis environment. These APIs provide a structured interface for robot control at the function-level, such as picking, placing, and alignment.

*Table 12.* Executable APIs in Genesis

| | End-effector Type | Template | Example |
|---|---|---|---|
| API | | Move Gripper To [Object] | `move_gripper_to(env, obj_name, pointing_to='down')` |
| | | Move To Position [Object] | `move_to_position(env, pos, pointing_to='down')` |
| | | Move Parallel [Object] | `move_parallel(env, direction, offset, ...)` |
| | | Pick [Object] | `pick(env, obj_name, pointing_to='down')` |
| | Parallel-jaw gripper | Place [Receptacle Object] | `place(env, obj_name, pointing_to='down')` |
| | | Rotate Gripper [Object] | `rotate_gripper(env, angle, steps=80)` |
| | | Grasp Handle [Object] | `grasp_handle(env, handle_name)` |
| | | Release Handle [Receptacle Object] | `release_handle(env)` |
| | | Open Gripper | `open_gripper(env)` |
| | | Close Gripper | `close_gripper(env)` |
| | | Move Gripper To [Object] | `move_gripper_to(env, obj_name, pointing_to='down')` |
| | | Move To Position [Object] | `move_to_position(env, pos, pointing_to='down')` |
| | | Move Parallel | `move_parallel(env, direction, offset, ...)` |
| | | Grasp Handle Robotiq85 [Object] | `grasp_handle_robotiq85(env, handle_name)` |
| | Robotiq 2F-85 gripper | Release Handle Robotiq85 [Receptacle Object] | `release_handle_robotiq85(env)` |
| | | Pick Robotiq85 [Object] | `pick_robotiq85(env, obj_name, pointing_to='down')` |
| | | Place Robotiq85 [Receptacle Object] | `place_robotiq85(env, obj_name, pointing_to='down')` |
| | | Rotate Gripper [Object] | `rotate_gripper(env, angle, steps=80)` |
| | | Open Robotiq85 | `open_robotiq85(env)` |
| | | Close Robotiq85 | `close_robotiq85(env)` |
| | | Move Gripper To [Object] | `move_gripper_to(env, obj_name, pointing_to='down')` |
| | | Move To Position [Object] | `move_to_position(env, pos, pointing_to='down')` |
| | | Move Parallel [Object] | `move_parallel(env, direction, offset, ...)` |
| | Vacuum gripper | Attach Vacuum Handle [Object] | `attach_vacuum_handle(env, handle_name)` |
| | | Detach Vacuum Handle [Receptacle Object] | `detach_vacuum_handle(env)` |
| | | Rotate Gripper [Object] | `rotate_gripper(env, angle, steps=80)` |
| | | Activate Vacuum | `activate_vacuum(env)` |
| | | Deactivate Vacuum | `deactivate_vacuum(env)` |

## C.3. Evaluation Metrics

We evaluate RECENT and all baselines using five metrics that capture both task performance and computational efficiency during skill grounding. Let $\mathcal{T} = \{t_1, t_2, \ldots, t_N\}$ denote the set of evaluation tasks, where each task $t_i$ consists of a sequence of subtasks $\{s_{i,1}, s_{i,2}, \ldots, s_{i,M_i}\}$.

**Success Rate (SR% ↑).** The task-level success rate measures the proportion of tasks that are fully completed. A task $t_i$ is considered successful if and only if all of its constituent subtasks are executed successfully:

$$\mathrm{SR} = \frac{1}{N} \sum_{i=1}^{N} \mathbf{1} \left[ \bigwedge_{j=1}^{M_i} \texttt{success}(s_{i,j}) \right], \tag{9}$$

where $\mathbf{1}[\cdot]$ is the indicator function and $\texttt{success}(s_{i,j})$ returns true if subtask $s_{i,j}$ is completed successfully.

**Goal-Conditioned Success Rate (GC% ↑).** The subtask-level success rate measures the proportion of individual subtasks that are completed successfully, providing a finer-grained measure of execution progress:

$$\mathrm{GC} = \frac{1}{\sum_{i=1}^{N} M_i} \sum_{i=1}^{N} \sum_{j=1}^{M_i} \mathbf{1} \left[ \texttt{success}(s_{i,j}) \right]. \tag{10}$$

**Grounding Overhead (GO% ↓).** The grounding overhead quantifies the additional computational cost incurred during skill adaptation relative to the original skill complexity. For each subtask $s_{i,j}$, let $\tau_{\mathrm{source}}(s_{i,j})$ denote the token count of the source skill code (excluding import statements), and let $\tau_{\mathrm{adapt}}(s_{i,j})$ denote the total number of tokens generated during repair and revise operations. The grounding overhead is computed as:

$$\mathbf{GO} = \frac{\text{\# tokens generated during grounding}}{\text{\# tokens in original skill code}} \times 100 \tag{11}$$

A lower GO indicates that the grounding process requires fewer token generations relative to the original skill size, reflecting more efficient adaptation through localized modifications rather than extensive code regeneration.

**Idle Time (IT ↓).** The idle time measures the cumulative duration of LM inference calls required for error correction during execution. For each task $t_i$, we sum the duration of all $\texttt{repair}$ and $\texttt{revise}$ operations:

$$\mathrm{IT} = \frac{1}{N} \sum_{i=1}^{N} \sum_{j=1}^{M_i} \sum_{k \in \mathcal{R}_{i,j}} d_k, \tag{12}$$

where $\mathcal{R}_{i,j}$ denotes the set of repair and revise LM calls for subtask $s_{i,j}$, and $d_k$ is the duration (in seconds) of call $k$. Lower IT indicates faster adaptation with reduced inference latency, which is critical for real-time embodied control.

**Execution Interruption Count (EI ↓).** The number of execution interruptions measures how frequently the agent must pause execution to perform error correction. For each task $t_i$, we count the number of LM calls with action type $\texttt{repair}$ or $\texttt{revise}$ across all subtasks:

$$\mathrm{EI} = \frac{1}{N} \sum_{i=1}^{N} \sum_{j=1}^{M_i} \left( c_{\mathrm{repair}}(s_{i,j}) + c_{\mathrm{revise}}(s_{i,j}) \right), \tag{13}$$

where $c_{\mathrm{repair}}(s_{i,j})$ and $c_{\mathrm{revise}}(s_{i,j})$ denote the number of repair and revise calls for subtask $s_{i,j}$, respectively. Lower EI indicates more stable execution with fewer runtime interventions. SR and GC are performance metrics where higher values indicate better task completion. GO, EI, and IT are efficiency metrics where lower values indicate more effective skill grounding. Together, these metrics capture the trade-off between task success and computational overhead that is central to efficient skill grounding with sLMs.

## C.4. Failure Categorization

Table 13 reports representative failure cases observed during skill execution and groups them by error type and underlying cause. The failures mainly arise from hallucinated variables or objects, hallucinated primitive API usage, and execution disruptions caused by motion planning or inverse kinematics failures. Hallucination errors occur when generated policy code refers to undefined variables or nonexistent scene objects, such as `variation_index`, `ball_position`, or `drawer`. Hallucinated API errors arise when the code invokes unsupported interfaces or incorrect arguments, for example calling `get_object_position` from the environment or passing `approach_distance` to `sawyer_pick()`. Other failures are caused by non-executable code, unreachable objects, blocked waypoints, workspace violations, or empty repair continuations. These errors show that deployment-time mismatches are not limited to syntactic mistakes, but also involve inconsistencies between execution bindings, robot capabilities, and environment constraints.

*Table 13.* Error Analysis and Classification

| Error Type | Error Analysis | Error Message |
|---|---|---|
| NameError | Hallucination | `name 'variation_index' is not defined`
`name 'ball_position' is not defined`
`name 'target_name' is not defined` |
| AttributeError | Hallucination
Hallucination (API) | `numpy.ndarray has no attribute 'index'`
`Environment has no attribute 'get_object_position'` |
| SkillFailure | Motion Planning & IK Failures | `WAYPOINT_FAIL – Blocked by another object.`
`WAYPOINT_FAIL – Waypoint too close to target center.`
`Alignment issues detected: Gripper misaligned with 'stand'` |
| | Non-executable Code | `Failed to get initial code: task.step() timed out` |
| PathOutOfWorkspace | Motion Planning & IK Failures | `Alignment issues detected: Gripper misaligned with 'tv_frame'.`
`All path planning strategies failed` |
| RuntimeError | Non-executable Code | `The call failed on the V-REP side. Return value: -1` |
| | Motion Planning & IK Failures | `Object 'green_cube' is not reachable`
`Object 'lemon' is not reachable)` |
| | Non-executable Code | `Revise returned empty continuation`
`Repair returned empty continuation` |
| TypeError | Hallucination (API) | `sawyer_pick() got unexpected keyword argument 'approach_distance'.` |
| | Hallucination | `'NoneType' object is not subscriptable`
`unsupported operand type(s) for +: 'NoneType' and 'float'`
`expected np.ndarray (got NoneType)` |
| ValueError | Hallucination | `Object 'drawer' not found in scene` |
| SyntaxError | Non-executable Code | `(' was never closed` |

## C.5. Implementation Details

All methods are evaluated under the same software and hardware configuration summarized in Table 14, using identical simulator setups, task definitions, primitive API specifications, and random seeds across five runs. For sLM-based methods, model execution is further constrained by the VRAM limits in Table 15, which emulates deployment settings with limited on-device memory. The 7B models are executed under a 12 GB memory budget using FP8 precision, while smaller models use FP16 precision within their corresponding memory budgets. These constraints ensure that performance differences mainly reflect the grounding procedure rather than differences in computational resources or evaluation conditions.

*Table 14.* Implementation details for evaluation

| Component | Specification |
| --- | --- |
| Python | 3.9.18 |
| CUDA / cuDNN | CUDA 12.4 / cuDNN 9.1 |
| PyTorch | 2.6.0 |
| Transformers | 4.53.1 |
| GPU | NVIDIA RTX 4090 (24 GB) |
| Random Seeds | 0, 1, 42, 123, 2024 (5 runs) |

*Table 15.* Small language model configurations under fixed memory budgets used in ablation studies

| Model | Family | Parameters | VRAM Limit | Precision |
| --- | --- | --- | --- | --- |
| Qwen2.5-Coder-1.5B | Qwen | 1.5B | 3.6 GB | FP16 |
| CodeGemma-2B | CodeGemma | 2.0B | 6 GB | FP16 |
| Qwen2.5-Coder-3B | Qwen | 3.0B | 7 GB | FP16 |
| Qwen2.5-Coder-7B | Qwen | 7.0B | 12 GB | FP8 |
| CodeGemma-7B | CodeGemma | 7.0B | 12 GB | FP8 |

# D. Additional Experimental Results

## D.1. Main experiment details

Tables 16, 17, 18, and 19 provide per-scenario breakdowns of the main experiments reported in Table 1. The aggregated results in Section 5.2 summarize overall performance across embodiment settings, and these detailed results show how each embodiment mismatch affects deployment-time grounding difficulty. The Panda → UR5 and Panda → Sawyer settings evaluate kinematic variation arising from differences in manipulator structure and motion constraints. The parallel-jaw gripper → vacuum gripper and parallel-jaw gripper → 2F-85 gripper settings evaluate end-effector variation under changes in grasping mechanisms and execution interfaces.

*Table 16.* Performance on continual embodied tasks under Kinematic variation **Panda → UR5**

| PANDA → UR5 | Kinematic variation | | | | |
|---|---|---|---|---|---|
| | SR (% ↑) | GC (% ↑) | GO (% ↓) | EI (count ↓) | IT (sec ↓) |
| CAP | 11.00±2.24 | 38.92±1.13 | 124.93±1.75 | 6.11±0.11 | 33.09±2.02 |
| CAP-CODEV-R1 | 33.00±12.55 | 50.00±7.94 | 100.24±53.52 | 3.26±1.95 | 11.48±2.86 |
| SCOT | 23.00±13.51 | 41.35±9.96 | 118.09±23.40 | 4.26±2.06 | 23.66±8.76 |
| PROGPROMPT | 25.00±0.00 | 49.95±1.36 | 112.16±18.28 | 3.59±1.27 | 47.33±30.42 |
| ROBOINSPECTOR | 40.00±3.54 | 60.35±6.62 | 101.94±1.84 | 0.74±0.17 | 27.79±1.39 |
| REPAIRAGENT | 50.00±3.54 | 67.03±6.01 | 28.95±24.45 | 1.65±0.30 | 42.84±36.09 |
| AGENTLESS | 24.00±8.22 | 45.95±4.27 | 138.08±9.20 | 1.42±0.99 | 48.83±20.04 |
| **RECENT (OURS)** | **75.00±0.00** | **82.43±1.35** | **11.81±6.00** | **0.70±0.30** | **2.47±2.02** |
| CAP-CODEX | 68.00±10.95 | 79.55±9.02 | 67.38±16.52 | 1.27±0.85 | 6.84±3.52 |

**Panda → UR5 kinematic variation**    As shown in Table 16, RECENT achieves the best task performance with 75.00% SR and 82.43% GC, outperforming the strongest baseline, RepairAgent, by 25.00 pp in SR and 15.40 pp in GC. It also reduces GO from 28.95% to 11.81%, EI from 1.65 to 0.70, and IT from 42.84 sec to 2.47 sec compared to RepairAgent. Compared to CaP-Codex, RECENT improves SR by 7.00 pp and GC by 2.88 pp, while reducing GO by 55.57 pp, EI by 0.57, and IT by 4.37 sec. These results show that localized refactoring of execution bindings can resolve kinematic embodiment gaps while preserving the functional structure of the skill.

*Table 17.* Performance on continual embodied tasks under Kinematic variation **Panda → Sawyer**

| PANDA → SAWYER | Kinematic variation | | | | |
|---|---|---|---|---|---|
| | SR (% ↑) | GC (% ↑) | GO (% ↓) | EI (count ↓) | IT (sec ↓) |
| CAP | 23.00±2.74 | 43.30±4.86 | 79.10±5.04 | 3.32±0.17 | 18.21±1.75 |
| CAP-CODEV-R1 | 7.00±10.37 | 22.20±19.64 | 118.53±45.33 | 5.00±3.03 | 23.51±11.02 |
| SCOT | 6.00±5.48 | 35.41±10.57 | 96.06±52.86 | 4.53±2.08 | 34.05±14.72 |
| PROGPROMPT | 38.00±9.08 | 57.03±9.67 | 120.90±47.41 | 3.03±1.58 | 14.18±6.18 |
| ROBOINSPECTOR | 42.00±5.70 | 61.32±6.49 | 256.86±5.68 | 1.55±0.15 | 92.17±2.09 |
| REPAIRAGENT | 34.00±10.25 | 61.08±5.27 | 79.08±52.16 | 1.82±0.49 | 58.22±10.54 |
| AGENTLESS | 21.00±18.17 | 39.46±20.05 | 238.09±101.34 | 1.77±1.01 | 53.80±31.03 |
| **RECENT (OURS)** | **71.00±7.42** | **81.08±3.02** | **3.89±2.58** | **0.74±0.41** | **1.08±1.52** |
| CAP-CODEX | 67.00±7.58 | 81.35±2.93 | 101.20±51.08 | 1.19±0.29 | 9.14±5.85 |

**Panda → Sawyer kinematic variation**    Table 17 further evaluates kinematic variation in the Panda → Sawyer setting, where RECENT obtains the highest SR of 71.00% and a GC of 81.08%. This corresponds to improvements of 29.00 pp in SR and 19.76 pp in GC over RoboInspector, the strongest baseline in this setting. RECENT also requires substantially lower grounding cost, reducing GO from 256.86% to 3.89%, EI from 1.55 to 0.74, and IT from 92.17 sec to 1.08 sec compared to RoboInspector. Against CaP-Codex, RECENT improves SR by 4.00 pp and reduces GO by 97.31 pp, EI by

0.45, and IT by 8.06 sec. These results indicate that localized refactoring can handle larger kinematic gaps without increasing deployment-time grounding overhead.

*Table 18.* Performance on continual embodied tasks under End-effector variation Panda with a **parallel-jaw gripper → vacuum gripper**

| PANDA | End-effector variation | | | | |
|---|---|---|---|---|---|
| *Parallel-jaw gripper →* *Vacuum gripper* | SR (% ↑) | GC (% ↑) | GO (% ↓) | EI (count ↓) | IT (sec ↓) |
| CAP | 16.67±0.00 | 32.21±5.48 | 80.01±0.00 | 3.04±0.01 | 72.47±0.56 |
| CAP-CODEV-R1 | 13.33±11.18 | 34.83±28.09 | 106.54±6.02 | 3.13±1.24 | 34.87±11.97 |
| SCOT | 13.33±7.45 | 47.62±6.88 | 108.02±55.07 | 2.06±0.34 | 40.87±11.12 |
| PROGPROMPT | 43.53±9.09 | 66.67±5.98 | 71.47±5.66 | 4.53±0.66 | 38.31±12.18 |
| ROBOINSPECTOR | 51.66±6.97 | 66.47±5.98 | 61.74±2.15 | 5.20±0.23 | 22.16±2.40 |
| REPAIRAGENT | 50.00±0.00 | 75.81±3.80 | 65.38±24.91 | 1.12±0.17 | 94.70±71.03 |
| AGENTLESS | 38.06±12.54 | 57.23±9.01 | 83.95±9.56 | 4.86±0.70 | 47.93±11.97 |
| **RECENT** (OURS) | **83.33±0.00** | **91.40±1.89** | **2.79±0.21** | **0.36±0.11** | **1.90±1.84** |
| CAP-CODEX | 71.67±4.56 | 87.31±2.15 | 38.12±10.67 | 0.62±0.12 | 10.64±4.62 |

**Parallel-jaw gripper → vacuum gripper end-effector variation**  Table 18 shows detailed results for the parallel-jaw gripper → vacuum gripper setting, where grasping behavior must be adapted from contact-based gripper control to suction-based manipulation. RECENT achieves the highest SR of 83.33% and GC of 91.40%, improving over the strongest baseline, RepairAgent, by 33.33 pp in SR and 15.59 pp in GC. It also reduces GO from 65.38% to 2.79%, EI from 1.12 to 0.36, and IT from 94.70 sec to 1.90 sec compared to RepairAgent. Compared to CaP-Codex, RECENT improves SR by 11.66 pp and GC by 4.09 pp, while reducing GO by 35.33 pp, EI by 0.26, and IT by 8.74 sec. These results show that execution-binding refactoring can efficiently adapt grasp and release primitives under changes in grasping mechanisms.

*Table 19.* Performance on continual embodied tasks under End-effector variation Panda with a **parallel-jaw gripper → 2F-85 gripper**

| PANDA | End-effector variation | | | | |
|---|---|---|---|---|---|
| *Parallel-jaw gripper →* *2F-85 gripper* | SR (% ↑) | GC (% ↑) | GO (% ↓) | EI (count ↓) | IT (sec ↓) |
| CAP | 27.27±5.08 | 40.45±11.88 | 78.61±4.89 | 3.33±0.76 | 38.78±1.94 |
| CAP-CODEV-R1 | 22.42±4.69 | 47.24±12.69 | 91.78±20.60 | 2.40±0.74 | 52.66±11.68 |
| SCOT | 19.44±5.89 | 62.19±4.10 | 52.89±41.82 | 1.63±0.06 | 51.78±6.73 |
| PROGPROMPT | 23.13±6.93 | 49.65±10.34 | 59.21±5.71 | 4.91±0.97 | 39.74±11.33 |
| ROBOINSPECTOR | 26.67±6.97 | 63.13±4.82 | 42.66±1.96 | 4.85±0.33 | 41.13±3.33 |
| REPAIRAGENT | 54.33±4.18 | 71.82±2.47 | 46.46±32.79 | 1.13±0.34 | 74.05±79.68 |
| AGENTLESS | 14.60±6.55 | 42.53±17.67 | 65.98±2.83 | 5.16±0.29 | 36.95±10.77 |
| **RECENT** (OURS) | **81.67±3.73** | **88.28±3.52** | **2.23±0.36** | **1.03±0.92** | **2.78±1.69** |
| CAP-CODEX | 78.03±7.23 | 88.31±1.62 | 45.27±9.17 | 0.59±0.05 | 9.94±4.64 |

**Parallel-jaw gripper → 2F-85 gripper end-effector variation**  The results in Table 19 correspond to the parallel-jaw gripper → 2F-85 gripper setting. Since both source and target embodiments use parallel grippers, the grounding challenge lies in adapting embodiment-specific primitive interfaces rather than changing the grasping mechanism itself. In this setting, RECENT achieves 81.67% SR and 88.28% GC, surpassing RepairAgent by 27.34 pp in SR and 16.46 pp in GC. The efficiency gain is more pronounced, with GO decreasing from 46.46% to 2.23% and IT decreasing from 74.05 sec to 2.78 sec compared to RepairAgent. Compared to CaP-Codex, RECENT achieves higher SR by 3.64 pp and reduces GO by 43.04 pp and IT by 7.16 sec. This indicates that preserving the skill structure and editing only embodiment-specific execution bindings is especially effective when the required adaptation is localized to gripper interfaces.

## D.2. Ablation experiment details

Table 20 and Table 21 provide detailed ablation results for Table 2 and Table 4, respectively.

*Table 20.* Ablation on model choice and sLM family

| FAMILY | MODEL SIZE | SR (% ↑) | GC (% ↑) | GO (% ↓) | EI (count ↓) | IT (sec ↓) |
|---|---|---|---|---|---|---|
| | 1.5B | 66.67 ±0.00 | 83.43 ±2.71 | 4.81 ±0.84 | 0.94 ± 0.03 | 7.83 ±0.88 |
| QWEN2.5-CODER | 3B | 83.33 ± 0.00 | 89.37 ± 0.12 | 3.42 ± 0.21 | 0.50 ± 0.02 | 15.24 ± 1.75 |
| | 7B | 83.33 ± 0.00 | 92.70 ± 0.90 | 2.64 ± 0.09 | 0.28 ± 0.01 | 2.62 ± 2.21 |
| CODEGEMMA | 2B | 66.67 ± 0.00 | 70.32 ± 1.85 | 6.38 ± 0.47 | 1.04 ± 0.21 | 1.21 ± 0.52 |
| | 7B | 83.33 ± 0.00 | 87.26 ± 0.54 | 3.22 ± 1.04 | 0.39 ± 0.23 | 19.40 ± 16.51 |

*Table 21.* Ablation on RECENT modules

| METHODS | SR (% ↑) | GC (% ↑) | GO (% ↓) | EI (count ↓) | IT (sec ↓) |
|---|---|---|---|---|---|
| W/O. ONTOLOGY GUIDANCE | 21.67 ±7.45 | 28.95 ±6.66 | 2.35 ±1.55 | 2.57 ±0.59 | 4.14 ±2.15 |
| → W/ API NAME SIMILARITY | 58.33 ±0.00 | 69.50 ±0.84 | 3.78 ±2.64 | 1.71 ±0.48 | 1.14 ±0.51 |
| W/O. IN-SITU ADAPTATION | 66.67 ±0.00 | 86.25 ±0.00 | 1.77 ±1.02 | 0.76 ±0.00 | 5.91 ±3.59 |
| → FULL RE-GENERATION | 26.67 ±9.13 | 44.53 ±5.78 | 89.63 ±1.40 | 2.44 ±0.44 | 91.21 ±2.50 |
| **RECENT** | **83.33** ± 0.00 | **91.40** ± 1.89 | **2.79** ± 0.21 | **0.36** ± 0.11 | **1.90** ± 1.84 |

## D.3. Real-world deployment

We include an additional real-world deployment study to assess whether RECENT can perform skill grounding on physical robot systems with heterogeneous embodiments. This study is designed to evaluate the practical applicability of our skill grounding procedure under realistic perception and manipulation conditions, complementing the simulation-based evaluations in Section 5 with evidence from physical robot deployments.

**Robot setup.** We evaluate our method on two heterogeneous real-world robot embodiments. The source embodiment is a 7-DoF Franka Research 3 (FR3) robotic arm equipped with a parallel two-finger gripper (Figure 8a), while the target embodiment is a 6-DoF UR7e arm equipped with a Robotiq 2F-85 parallel gripper (Figure 8b). Beyond differences in kinematics and end-effector hardware, the two platforms also differ in how their primitives are exposed and composed. While some primitives are shared across platforms, the FR3 implementation encapsulates object detection within guarded pick primitives to maintain a safer abstraction boundary around low-level motion control, whereas the UR7e implementation exposes detection and picking as separate primitives, using a modular interface that accepts detected object poses explicitly in picking or motion-level calls. Each robot operates in a tabletop workspace with a similar task layout. For perception, each robot is equipped with an Intel RealSense D435 RGB-D camera mounted in an eye-in-hand configuration on the end-effector. The captured RGB-D observations are processed using SAM 3 (Carion et al., 2026) for class-agnostic segmentation, followed by AnyGrasp (Fang et al., 2023) to estimate feasible grasp poses for detected objects.

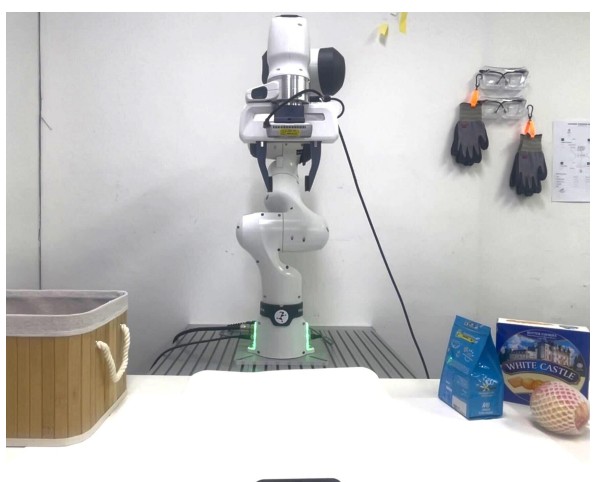 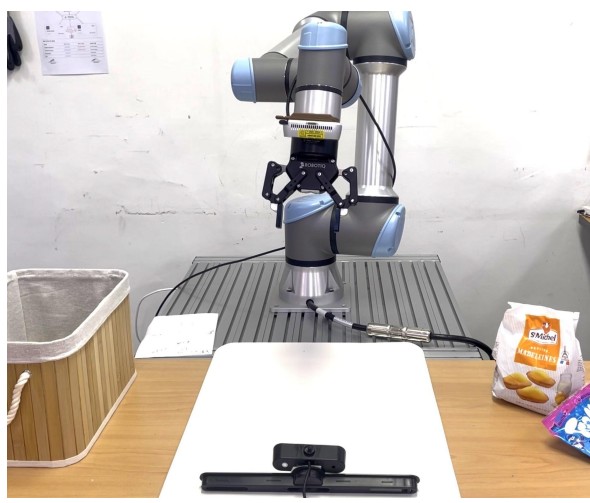

*(a)* FR3 with a two-finger gripper (source embodiment)          *(b)* UR7e with a Robotiq 2F-85 gripper (target embodiment)

*Figure 8.* Real-world robot platforms for deployment. The embodiments differ in robot morphology and gripper hardware.

**Safety safeguards for real-robot execution.** RECENT performs code refactoring at the skill level and does not directly bypass low-level robot safety mechanisms. Ontology-based unit tests validate not only API-level compatibility but also embodiment-level execution consistency, so unsafe or infeasible patched behaviors can be treated as validation failures and fed back into the refactoring loop. At the control level, patched skill code is executed through safeguarded robot controllers and motion planners, including configured workspace bounds, joint-limit checks, and collision-aware trajectory validation. In our implementation, collision-aware motion planning is handled with cuRobo (Sundaralingam et al., 2023) when available, while hardware-level safety limits and collision halting are enforced by the robot execution stack.

**Scan-and-bag task.** We evaluate RECENT on a real-world scan-and-bag task in a checkout-counter setting, where the robot sequentially scans checkout items placed on the counter using a barcode scanner and places the scanned items into a bagging basket. Figure 9 shows the real-world checkout-counter setup, where the unscanned area, the scanning station, the barcode scanner, and the bagging basket are annotated. In this task, a human places checkout items in the unscanned area, and the robot detects the items, estimates feasible grasp points, picks one item, moves it through the scanning station to scan its barcode with the tabletop barcode scanner, and places the scanned item into the basket located next to the scanning station. A trial is considered successful only when the robot completes this process for all five checkout items placed on the checkout counter. As illustrated in Figure 10 and Figure 11, each trial consists of five repeated item-level subgoals, where

the robot sequentially picks, scans, and bags checkout items. Each subgoal corresponds to the complete pick-scan-place sequence for a single target item.

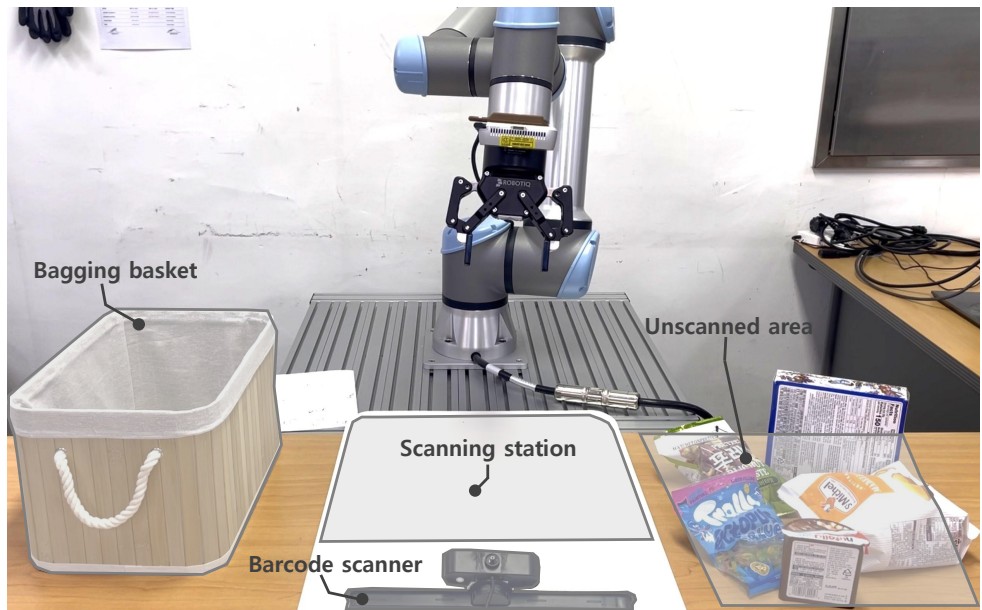

*Figure 9.* Real-world checkout-counter setup for the scan-and-bag task. The unscanned area, where items awaiting scanning are placed, scanning station, barcode scanner, and bagging basket are annotated in the scene.

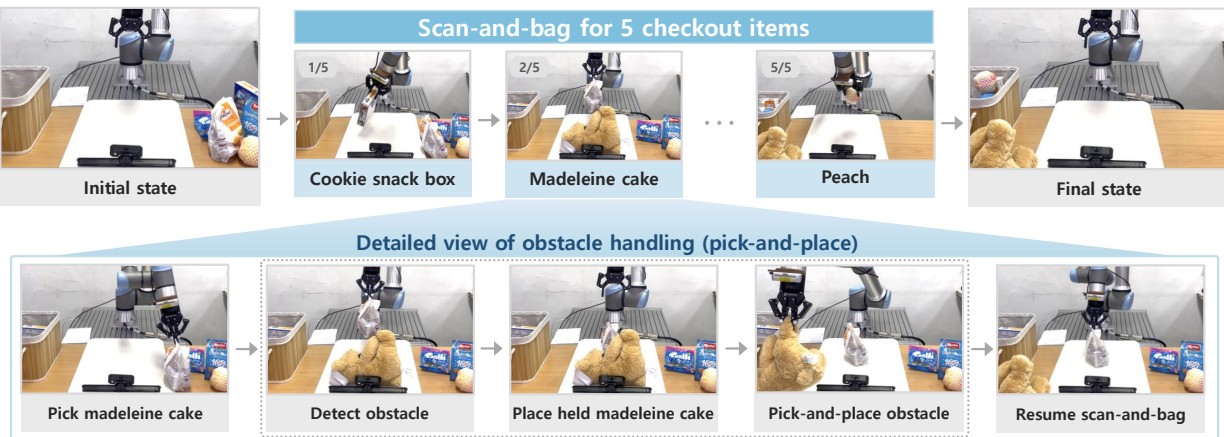

*Figure 10.* Task sequence for the real-world scan-and-bag task with pick-and-place obstacle handling. The label $N/5$ indicates that the robot has completed scan-and-bag for $N$ out of five checkout items. The second row provides a detailed view of the obstacle-handling condition during the scan-and-bag sequence for the second item. After picking the second item, the madeleine cake pouch, a graspable teddy bear is introduced into the scanning station. The robot handles this obstacle using the pick-and-place removal branch, which is already available in the source skill, and then resumes the original scan-and-bag sequence.

**Obstacle-handling condition.** During the task, an unexpected obstacle may be introduced into the scanning station and block the barcode scanner, for example when a checkout item is mistakenly placed in front of the scanner. Before executing the scan step, the robot checks whether the scanning station is occupied by an unexpected obstacle using the RGB-D observation. If the scanning station is blocked, the robot must remove the obstacle from the scanning station to the disposal area and then resume the original scan-and-bag sequence. When the robot is already holding a target item, it first places the item at a temporary location, removes the obstacle, and then continues the task.

An obstacle is considered graspable if AnyGrasp provides a stable top-grasp pose for pick-and-place removal, and non-graspable if its shape or placement makes stable grasping unreliable, requiring a sweeping motion instead. The obstacle-

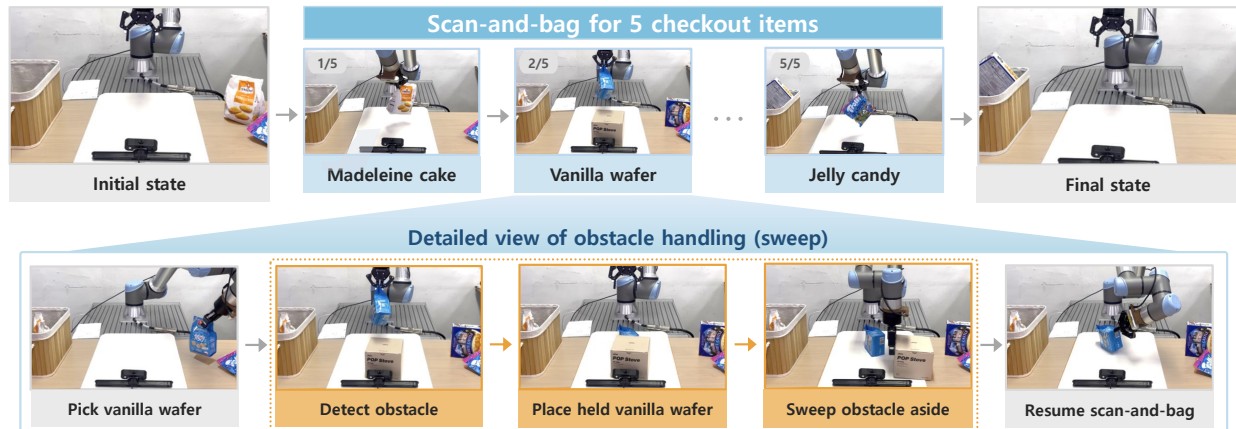

*Figure 11.* Task sequence for the real-world scan-and-bag task with sweeping obstacle handling. The label $N/5$ indicates that scan-and-bag has been completed for $N$ out of five checkout items. The second row details the obstacle-handling condition during scan-and-bag for the second item. After picking the second item, the vanilla wafer pouch, a non-graspable camping stove box is introduced into the scanning station. Since this obstacle cannot be reliably removed by pick-and-place, a sweeping branch is patched into the obstacle-handling routine during execution, shown in the orange highlighted region, and the original scan-and-bag sequence is then resumed.

handling routine is already included in the source skill. In the source setting, all obstacles are graspable and can therefore be removed using only the pick-and-place branch. In the target real-world setting, some obstacles are non-graspable, requiring the robot to select either pick-and-place or sweeping depending on the graspability of the observed obstacle. Thus, this task is designed not merely as a real-world execution demo, but as an environment-dependent adaptation setting in which RECENT must locally reground the obstacle-removal behavior based on execution-time observations.

**Trial configuration.** We conduct 10 trials with trial-level variations in the item sets and obstacle-handling conditions. Each trial corresponds to a complete scan-and-bag episode in which the robot processes all five checkout items placed on the checkout counter. Among the 10 trials, 8 include an obstacle-handling condition, while the remaining 2 are conducted without obstacle introduction to evaluate nominal scan-and-bag execution. For each trial, the obstacle-handling condition is defined by whether an obstacle is introduced, when it occurs within the task sequence, the type of obstacle, and whether the obstacle is graspable. The obstacle introduction timing is specified with respect to the task phase, such as after picking an item, before scanning, or after scanning. Human intervention is limited to placing the checkout items and introducing obstacles according to the trial configuration, while robot execution and recovery are performed autonomously. Table 22 summarizes the trial-level configuration, including the item set assignment and the obstacle-handling condition for each trial. The corresponding item sets are detailed in Table 23.

*Table 22.* Trial configurations for the real-world scan-and-bag task. Each trial corresponds to a complete episode in which the robot processes five checkout items.

| Trial | Item Set | Interruption | Timing | Obstacle | Graspable | Removal Strategy |
|---|---|---|---|---|---|---|
| 1 | A | No | – | – | – | Nominal |
| 2 | B | Yes | After pick | Teddy bear | Yes | Pick-and-place |
| 3 | B | Yes | After pick | Camping stove box | No | Sweep |
| 4 | B | Yes | After pick | Toolbox | No | Sweep |
| 5 | A | Yes | After scan | Camping stove box | No | Sweep |
| 6 | B | No | – | – | – | Nominal |
| 7 | A | Yes | After scan | Teddy bear | Yes | Pick-and-place |
| 8 | B | Yes | Before scan | Toolbox | No | Sweep |
| 9 | B | Yes | After pick | Teddy bear | Yes | Pick-and-place |
| 10 | A | Yes | Before scan | Camping stove box | No | Sweep |

*Table 23*. Item set definitions used in the real-world scan-and-bag trials. Each item set contains five checkout items placed on the checkout counter.

| Item Set | Checkout Items |
|:---:|---|
| A | jelly candy pouch, pretzel snack pouch, cookie snack box, madeleine cake pouch, chocolate dip-and-stick snack pack |
| B | madeleine cake pouch, jelly candy pouch, vanilla wafer pouch, cookie snack box, mesh-wrapped peach |

**Evaluation results.** Table 24 reports the real-world scan-and-bag results over 10 trials. RECENT successfully completes 8 out of 10 trials, whereas CaP completes 4 out of 10 trials. Grounding overhead is reduced from 125.80% with CaP to 15.70% with RECENT, and execution interruptions decrease from 3.00 to 0.40 per trial. The average interruption time is also reduced from 23.51s with CaP to 2.34s with RECENT. The two failed trials occur when an obstacle is introduced immediately before the scan motion, leaving insufficient time and no valid yet-to-be-executed recovery span for localized patching before the scanner interaction.

*Table 24*. Performance comparison on the real-world scan-and-bag task over 10 trials.

| METHOD | SUCCESSES / TRIALS ($\uparrow$) | GC (% $\uparrow$) | GO (% $\downarrow$) | EI (count $\downarrow$) | IT (sec $\downarrow$) |
|---|:---:|:---:|:---:|:---:|:---:|
| CAP | 4/10 | $80.00 \pm 17.80$ | $125.80 \pm 21.94$ | $3.00 \pm 1.45$ | $23.51 \pm 13.12$ |
| RECENT | 8/10 | $96.00 \pm 8.00$ | $15.70 \pm 2.60$ | $0.40 \pm 0.49$ | $2.34 \pm 0.63$ |

The main difference comes from how the two methods handle obstacle-induced execution-time mismatches. When the scanning station is occupied by a non-graspable obstacle, RECENT does not wait for a failed grasp attempt to trigger full recovery. Instead, the unit-test-based validation detects the graspability violation before the corresponding obstacle-removal code is executed, and RECENT patches only the affected, yet-to-be-executed code span by replacing the grasp-based removal branch with a sweep-based removal branch. This anticipatory localized patching allows the robot to clear the scanning station and resume the scan-and-bag sequence with low interruption time and low grounding overhead. In contrast, CaP handles the same mismatch through policy regeneration. This makes recovery less stable, as the regenerated program may fail to produce a consistent obstacle-handling branch, and even successful recovery often requires longer execution pauses because unrelated parts of the task program are regenerated together with the obstacle-handling logic.

**D.4. Comparison with deployment-time learning baselines**

We compare RECENT with learning-based baselines operating at deployment time, while our main experiments primarily focus on baselines centered on code generation and program repair. This comparison examines a different deployment-time adaptation strategy, where reference samples are used to adapt the model or prompt to the target task. We find that such learning-based adaptation can improve task success, but still incurs substantial grounding cost, measured by GO and Initial Grounding Time (IGT), and remains below RECENT in success rate. In contrast, RECENT shifts much of the adaptation cost from deployment-time to the offline stage. This is enabled by its offline-validated skill repository and localized execution-binding refactoring, which reduce deployment-time cost while achieving higher task success.

**Deployment-time learning baselines.** We consider two learning-based baselines that use reference samples at deployment time. Test-Time Training (TTT) (Akyürek et al., 2025) retrieves top-$k$ examples and trains a task-specific LoRA module before generating the initial executable code. When execution requires code regeneration, we train a separate LoRA module for the failure-conditioned regeneration objective and use it to produce the corrected code. Efficient Prompt Retriever (EPR) (Rubin et al., 2022) retrieves top-$k$ examples and directly uses them as in-context demonstrations for both initial code generation and corrected code generation. Both baselines require deployment-time reference samples, where $k$ denotes the number of retrieved examples. We evaluate TTT with $k = 2$ and $k = 10$, and EPR with $k = 2$, using deterministic oracle retrieval results for both baselines to isolate the effect of deployment-time adaptation from retrieval errors.

**Evaluation metrics.** We report the same metrics used in the main experiments, namely SR, GC, GO, EI, and IT, and additionally include IGT for this comparison. IGT measures the wall-clock time from receiving a target task to obtaining the first executable code, including any deployment-time adaptation required before execution. This additional metric captures the pre-execution latency introduced by deployment-time learning baselines, such as task-specific LoRA updates or retrieval-conditioned prompting, which is not reflected in IT because IT only measures robot interruption time during execution.

**Evaluation results.** Table 25 shows the comparison with deployment-time learning baselines. RECENT achieves a 15.78 pp higher SR than TTT with $k = 10$ and reduces IGT by 6.49× compared with EPR with $k = 2$. This is because TTT and EPR perform task-level grounding at deployment time, increasing adaptation cost. Even with oracle retrieval, they reconstruct task-level executable policy code from retrieved samples, causing not only execution bindings but also functional logic to be regenerated together. RECENT stabilizes semantic intent in advance through the offline-validated skill repository and only locally edits embodiment- and environment-specific execution bindings at deployment time, thereby achieving more stable and successful grounding with lower overhead.

*Table 25.* Performance comparison with deployment-time learning baselines.

| METHOD | SR (% ↑) | GC (% ↑) | GO (% ↓) | EI (count ↓) | IT (sec ↓) | IGT (sec ↓) |
|---|---|---|---|---|---|---|
| TTT ($k = 2$) | $57.72 \pm 0.80$ | $65.31 \pm 1.62$ | $61.17 \pm 3.93$ | $1.87 \pm 0.07$ | $73.94 \pm 4.50$ | $46.06 \pm 0.50$ |
| TTT ($k = 10$) | $66.16 \pm 8.59$ | $74.60 \pm 8.31$ | $75.83 \pm 6.86$ | $1.62 \pm 0.13$ | $69.31 \pm 7.15$ | $64.24 \pm 0.15$ |
| EPR ($k = 2$) | $59.72 \pm 2.41$ | $74.58 \pm 0.96$ | $49.30 \pm 18.84$ | $1.14 \pm 0.07$ | $71.62 \pm 10.94$ | $12.86 \pm 0.15$ |
| RECENT(ours) | $\mathbf{81.94 \pm 2.41}$ | $\mathbf{90.80 \pm 2.22}$ | $\mathbf{2.57 \pm 0.03}$ | $\mathbf{0.42 \pm 0.14}$ | $\mathbf{2.60 \pm 0.74}$ | $\mathbf{1.98 \pm 0.03}$ |

