# OpenReview forum: "Efficient Skill Grounding via Code Refactoring with Small Language Models"
_ICML.cc/2026/Conference — ICML 2026 regular_

### Official Review · Reviewer_r6ER · 2026-02-14

**Soundness:** 3
**Presentation:** 2
**Significance:** 2
**Originality:** 3
**Overall Recommendation:** 4
**Confidence:** 3

**Summary:**

The paper studies skill grounding for embodied agents when you cannot rely on a large cloud LLM at run time. It proposes RECENT, which stores skills as executable code and splits each skill into stable task intent and robot/environment-specific bindings.

RECENT edits only the binding parts using Fill-in-the-Middle code refactoring, instead of regenerating whole programs. On long-horizon manipulation tasks with kinematic and end-effector variation, RECENT reports higher success and much lower grounding overhead and interruptions than several sLM baselines.

**Compliance With Llm Reviewing Policy:**

Affirmed.

**Final Justification:**

Given the clarification on initialization cost and overall positioning, I believe the paper is acceptable for publication.

**Key Questions For Authors:**

Considering recent advancements in the embodied planning and execution field, where do you position the task of Skill Grounding, is it a sub-area that serves (m)LLM based embodied task planning? How does it related to other methods of embodied task execution such as VLA and other real-time robotic policies?

**Limitations:**

yes

**Strengths And Weaknesses:**

Strength:

The refactoring mechanism is well specified as localized infilling targets plus hints. The evaluation uses success metrics and also efficiency metrics that match the goal of on-device grounding.

The novelty is not “a new policy.” It is a system-level reframing: treat grounding as local code refactoring plus ontology-guided target selection. This may be useful for code-based robot control.

Weakness:

A lot of manual design efforts go into designing recent. it heavily rely on an ontology and a validated offline skill library. this may hide large setup cost. The paper shows an ablation where removing offline validation hurts performance, which supports the design, but also suggests a strong dependency on that offline stage.

I still wanted more detail on the ontology itself. For example, how you build it, how large it is, and how often you need manual edits.

The framing is easy to follow, but I want more information on how the authors actually positon this work and the field within recent advances in embodied planning and execution. See my question below.

---

> ### Author Rebuttal · Authors · 2026-03-31
>
> We appreciate the reviewer's balanced assessment and clarify the points below.
>
> > **W1.** Offline validation cost and dependency.
>
> We agree that RECENT introduces an upfront setup cost in constructing the skill ontology and the offline validated skill repository. However, this is a one-time investment in specifying reusable skill semantics, rather than per-task engineering of implementation detail.
> In RECENT, these semantics are captured through a skill ontology [1] as capabilities, preconditions and expected effects, which are generally more stable across robot embodiments than execution-specific APIs, parameters, and bindings.
>
> Based on this abstraction, we generate executable skills for a general-purpose source embodiment (e.g., Franka Emika Panda) using a cloud-scale LLM in a Turing-complete representation (e.g., Python).
> This representation is modular and separates semantic intent from execution bindings.
> As a result, the offline stage produces a reusable executable prior that can be efficiently adapted through localized updates at deployment time.
> During offline validation, we check whether observed execution outcomes satisfy the ontology-defined conditions.
>
> Engineering effort is therefore concentrated on building an automated pipeline rather than manually crafting individual skills. This supports scalable construction of a shared skill library that can be reused across diverse robot embodiments with minimal adaptation, amortizing the initial setup cost over many deployment scenarios.
>
> As shown in Table 3 of the main paper, offline semantic validation reduces the reasoning burden on sLMs during deployment, rather than simply reflecting dependence on the offline skill repository.
>
> To evaluate deployment-time efficiency, we additionally compare against two learning-based baselines, test-time training (TTT) [2] and a few-shot prompt retrieval method (EPR) [3], and measure efficiency using initial grounding time (IGT), the time required to produce an executable skill.
>
> |Methods|SR↑|GO↓|IGT↓|
> |-|-|-|-|
> |TTT (k=2)|57.72±0.80|61.17±3.93|46.06±0.50|
> |TTT (k=10)|66.16±8.59|75.83±6.86|64.24±0.15|
> |EPR (k=2)|59.72±2.41|49.30±18.84|12.86±0.15|
> |RECENT|81.94±2.41|2.57±0.03|1.98±0.03|
>
> RECENT achieves the highest success rate while requiring the least initial grounding time. Compared with TTT (k=10), it improves SR by 15.78%, and compared with EPR, it reduces IGT by 6.45×, showing reduced deployment-time grounding cost.
>
> > **W2.** Ontology details.
>
> To clarify this point, we provide details on the ontology’s construction, scale, and manual effort. The ontology follows a KnowRob-based representation [1], where robot knowledge is organized as a structured semantic representation rather than manually engineered per task.
> In our implementation, each robot is annotated once with its embodiment profile and primitive API metadata, and cross-robot relations are instantiated through deterministic alignment rules within a shared triple-based ontology.
>
> In terms of scale, the ontology remains reusable. Across all experiments, it comprises 5 robots, 40 primitives (82 parameters, 67 preconditions, and 66 effects), 29 capabilities, and 40 instantiated substitution/equivalence relations, totaling 1,232 triples. This reflects a shared abstraction layer, not a task-specific knowledge base.
>
> The manual effort is limited to aligning URDF/SRDF naming and defining primitive APIs. Once these are defined, the ontology is automatically constructed in approximately 0.08 seconds via a deterministic Python pipeline that builds the graph and derives capability and substitution relations.
>
> In our experiments, new skills did not require manual ontology edits, as they are generated and validated automatically within this framework.
>
> > **W3 Q1.** Positioning.
>
> We position our work within robotic language grounding [4], which spans a spectrum from symbolic representations to embedding-based policies.
> Model-as-policy approaches (i.e., VLA) lie on the embedding side, where skill grounding is implicit in learned representations. In such approaches, functional intent (what is true) and execution details (how to perform) are entangled within model parameters, making adaptation to new embodiments rely on retraining or fine-tuning.
>
> By contrast, code-as-policy approaches lie on the symbolic side, where grounding can be performed over structured, interpretable representations. Building on this perspective, we focus on a previously underexplored deployment-time grounding of symbolic skills. We represent skills as codes that separate functional specifications from embodiment- and environment-specific execution bindings, and cast grounding as a localized refactoring problem.
>
> [1] Representations for robot knowledge in the knowrob framework
>
> [2] The Surprising Effectiveness of Test-Time Training For Few-Shot Learning
>
> [3] Learning to Retrieve Prompts for In-Context Learning
>
> [4] A Survey of Robotic Language Grounding: Tradeoffs between Symbols and Embeddings

---

> > ### Author Rebuttal · Reviewer_r6ER · 2026-04-05
> >
> > Thank you for the detailed rebuttal, I will raise my score.

---

> > > ### Author Response · Authors · 2026-04-06
> > >
> > > We appreciate your consideration of our rebuttal and your positive update.
> > >
> > > We are glad that our clarifications on the role of the offline validation stage, the construction and scalability of the ontology, and the positioning of skill grounding have fully addressed your concerns.
> > >
> > > Thank you again for your careful reading and evaluation.
> > >
> > > Sincerely,
> > >
> > > The Authors

---

### Official Review · Reviewer_fSaj · 2026-03-11

**Soundness:** 3
**Presentation:** 2
**Significance:** 3
**Originality:** 2
**Overall Recommendation:** 3
**Confidence:** 2

**Summary:**

The paper focuses on enabling sLMs to perform skill grounding efficiently in embodied agents. RECENT consists of three stages: first, an offline skill repository; second, ontology-based reasoning; and third, in-situ adaptation. Through these three stages, the method allows sLMs to reuse skills via localized edits rather than regenerating them from scratch.

**Compliance With Llm Reviewing Policy:**

Affirmed.

**Key Questions For Authors:**

- Have the authors considered evaluating the method on real robots? This would help verify its generalization ability and scalability.

- The novelty of the paper could be stronger. RECENT appears more like a strong system integration effort that combines several existing ideas into a unified framework, making the contribution somewhat more engineering-oriented. The authors could further clarify the paper’s novelty.

- Have the authors analyzed failure cases of localized refactoring? In what types of tasks or embodiment gaps does localized editing break down, making full regeneration necessary?

- The authors should revise several textual errors and incorrect references in the paper.

**Limitations:**

yes

**Strengths And Weaknesses:**

**Strengths:**

- The paper addresses a very practical issue in embodied agents: skills often cannot be directly reused across different robots and environments, while sLMs are not good at regenerating entire code sequences from scratch. The proposed RECENT framework provides a meaningful improvement on this problem.

- The overall experimental design is fairly complete, and the method demonstrates strong performance gains in the experiments. The ablation studies are also convincing and provide support for the authors’ claims.

- The paper is generally clearly written and easy to follow.

**Weaknesses:**

- In terms of novelty, the paper’s innovation could be stronger. RECENT feels more like a strong system integration effort that combines several existing ideas into a unified framework, making the contribution somewhat more engineering-oriented.

- Although the experimental results are strong, the evaluation is still limited to simulated environments.

- The writing is not fully rigorous in some details. For example, the ablation on offline validation is presented in Table 3, but the main text incorrectly states that “Table 4 shows the effect of offline skill validation.”

---

> ### Author Rebuttal · Authors · 2026-03-31
>
> We thank the reviewer for highlighting the practical aspects that motivated our design, and we address each concern below.
>
> > **W1 Q2.** Novelty beyond system integration.
>
> The key novelty of our work goes beyond the integration of existing techniques and lies in the reformulation of skill grounding into a form that is tractable for sLMs.
> In prior approaches, grounding is typically performed via full policy relearning or regeneration, requiring joint adaptation over both task semantics and context-specific execution details, which is inherently difficult for capacity-limited models.
>
> RECENT introduces a structured decomposition of skill grounding through two complementary separations:
> (1) separating invariant semantic intent from execution-specific bindings within the skill representation and
> (2) distinguishing determined (embodiment-resolvable) factors from undetermined (environment-contingent) factors.
>
> Representing skills as executable code provides a modular and compositional structure with explicit symbolic interfaces, making the separation between intent and bindings an explicit representational property.
>
> The separation also distinguishes what is fully determined by the robot and task specification from what remains underspecified until execution. Grounding then becomes constrained, localized updates for the determined parts, with the remaining uncertainty resolved only when execution context becomes available, keeping the overall adaptation within operations that small language models can handle reliably.
>
> While we leverage recent advances in code-based robotic programming, the novelty lies in restructuring the problem so that skill grounding becomes feasible under limited model capacity and deployment constraints.
>
> > **W2 Q1.** Real-robot evaluation.
>
> Beyond the simulated benchmarks in the submission, we also conducted experiments on real robot hardware, indicating that RECENT remains effective under real-world execution constraints.
> We constructed the offline skill repository on a Franka Research 3 and deployed it on a UR7e to evaluate deployment-time grounding under an embodiment gap. We considered a retail "scan-and-bag" scenario requiring coordinated manipulation and scanner interaction, with occlusions and physical disturbances near the scanner that required online adaptation.
>
> We compared against CaP across 10 trials. Idle Time (IT) denotes the cumulative inference time for adaptation during execution.
>
> |Methods|# Successful Runs↑|GC↑|GO↓|EI↓|IT↓|
> |-|-|-|-|-|-|
> |CaP|4/10|80.00±17.80|125.80±21.94|3.00±1.45|32.59±13.37|
> |RECENT|8/10|96.00±8.00|15.70±2.60|0.40±0.49|0.49±0.70|
>
> CaP incurs high IT due to repeated full regeneration, whereas RECENT maintains low IT through localized in-situ patching.
>
> We will include these real-robot results in the final version, along with setup photos, hardware details, and additional real-world scenarios.
>
> > **W3 Q4.** Writing issues.
>
> We will correct the sentence 'Table 4 shows the effect ...' to 'Table 3 shows the effect ...' in the final version. We will also revise 'five evaluation settings' to 'four evaluation settings'.
>
> > **Q3.** Limits of refactoring.
>
> Localized refactoring may fail during in-situ adaptation when the mismatch cannot be resolved by skill grounding alone and the skill must change due to a shift in state (e.g., object slip causing grasp failure). In such cases, full regeneration or switching to a different skill may become necessary. This is closely related to the classic distinction between plan repair and replanning [1]. Such cases fall into two categories:
>
> (1) Ontology-resolvable structural adaptations: Some embodiment mismatches require restructuring the skill logic (e.g., adding a short preparatory step or splitting one coarse skill into multiple primitives), but remain expressible using existing ontology primitives.
> RECENT handles these cases via ontology-guided refactoring, restructuring the failing code region into a finer-grained primitive sequence until it passes execution-time validation.
>
> (2) Ontology-level capability extension: These arise when the required capabilities are not represented in the ontology. For example, transferring a manipulation skill from a fixed-base manipulator to a legged mobile manipulator requires introducing navigation-related capabilities (e.g., locomotion and path planning).
> Once such extensions are provided, RECENT can operate on the updated ontology, but autonomously expanding the ontology at deployment time is out of scope and is better viewed as an incremental learning problem [2].
>
> We emphasize that RECENT operates under a given ontology and does not introduce new skill capabilities at deployment time. Extending the framework to handle capability expansion is an important direction for future work toward more compositional generalization scenarios.
>
> [1] Plan Stability: Replanning versus Plan Repair
>
> [2] Incremental Learning of Retrievable Skills For Efficient Continual Task Adaptation

---

> > ### Author Rebuttal · Reviewer_fSaj · 2026-04-03
> >
> > Thank the authors for the rebuttal; it has addressed most of my concerns. I will raise my score.

---

> > > ### Author Response · Authors · 2026-04-03
> > >
> > > We are happy to see that most concerns have been resolved, and we also appreciate your considering a more positive assessment than the original score of 3. If there are any remaining concerns, we would be grateful for the opportunity to discuss them further.
> > >
> > > Thank you again for your time and consideration.
> > >
> > > Best regards,
> > >
> > > The Authors

---

### Official Review · Reviewer_Tqg9 · 2026-03-11

**Soundness:** 3
**Presentation:** 3
**Significance:** 3
**Originality:** 3
**Overall Recommendation:** 4
**Confidence:** 3

**Summary:**

This paper presents RECENT, a refactoring-centric framework for skill grounding in embodied agents using small language models (sLMs). The central idea is that when transferring a skill from one robot to another, most of the task logic remains unchanged - only the execution-specific bindings (API calls, parameters, sensing interfaces) need updating. RECENT formalizes this through a skill ontology encoding capabilities, requirements, and semantic relations between robots, and decomposes deployment-time mismatches into "determined" factors (embodiment gaps resolvable before execution via ontology-guided substitution) and "undetermined" factors (environmental variations handled at runtime through anticipatory unit tests). Both types of adaptation use sLM-based Fill-in-the-Middle (FIM) infilling for localized code edits rather than full program regeneration. The framework also relies on an offline repository of LLM-generated, validated skill code on a reference robot (Franka Panda). Experiments in CoppeliaSim and Genesis cover kinematic transfer (Panda to UR5/Sawyer) and end-effector transfer (parallel-jaw to vacuum/2F-85) on long-horizon manipulation tasks requiring up to 54 sequential API calls. RECENT with Qwen2.5-Coder-7B substantially outperforms eight baselines spanning CaP variants, embodied code agents, and automated program repair agents, while matching or slightly exceeding the SR of an LLM-based CaP variant using GPT-5.2-Codex.

**Compliance With Llm Reviewing Policy:**

Affirmed.

**Key Questions For Authors:**

1. How are the ontology's substitution relations (substitutable-by, functionalEquivalent) constructed? Are they hand-authored per robot pair, rule-derived, or partially learned? How many relations exist for the tested scenarios, and what effort is needed to extend to a new robot?

2. The Inference Time (IT) metric is defined in Appendix C.3 and reported in the appendix breakdown tables, but absent from the main Table 1. Can you include IT in Table 1 alongside GO? This would substantially strengthen the efficiency comparison.

3. Could you add a baseline that uses FIM-based localized editing without ontology-guided substitution hints (e.g., using API name matching or call-graph diffs to identify edit locations)? This would disentangle the ontology's contribution from the localized editing strategy.

4. What safeguards exist against in-situ patches that produce unsafe motions? Is there any trajectory validation, IK feasibility check, or sandboxing before executing patched code?

5. Can you provide significance tests or confidence intervals for the key pairwise comparisons in Table 1? With 3 seeds and the observed variance, some gaps may not be statistically reliable.

6. In Table 4, removing in-situ adaptation costs only 8.33% SR. Which tasks or scenarios benefit most from this component, and is this difference significant?

**Limitations:**

The limitations section acknowledges that the ontology is assumed stable and that skill variants may diverge in real deployments. This is fair but misses the bigger issues. The simulation-only evaluation is not discussed as a limitation despite the paper repeatedly emphasizing deployment constraints and on-device execution. Safety implications of runtime code patching are absent entirely. The efficiency narrative relies on GO (a token-count ratio that favors the proposed method by design) while the paper's own IT metric is confined to appendix tables and absent from the main results. And the manual effort and domain knowledge required to build the ontology - which is the dominant factor in the ablation - deserves explicit discussion as a scalability concern.

**Strengths And Weaknesses:**

**Strengths:**

The refactoring-vs-regeneration framing is the paper's core insight and I think it is a good one. When porting a skill between robots, the high-level task structure (pick, move, place) stays the same - what changes is which API to call, what parameters to pass, and how to handle sensing differences. Existing code-as-policies approaches regenerate the entire program, which wastes tokens and gives the sLM unnecessary opportunities to corrupt task logic that was already correct. Using FIM for localized edits instead is well aligned with how sLMs actually perform - constrained infilling is a much easier task than open-ended generation at the 7B scale. The determined/undetermined decomposition is practical too: embodiment mismatches like "this robot has a vacuum gripper instead of parallel jaws" can be fully resolved before execution using the ontology, while environmental factors like contact dynamics or partial observability genuinely cannot. Splitting these into pre-execution and runtime phases is sensible.

The experimental setup is thorough. Eight baselines across three categories (CaP variants including a distilled sLM and GPT-5.2-Codex, agentic programming with ProgPrompt and RoboInspector, and automated repair with RepairAgent and Agentless), all sharing the same reference skills and evaluated under identical conditions. The component ablation in Table 4 is the most informative result in the paper: removing the ontology collapses SR from 83.33% to 19.44%, while removing in-situ adaptation costs only 8.33%. This kind of honest reporting, where one component clearly dominates, is more useful than ablations that spread credit evenly. The sLM scale study (Table 2) across Qwen2.5-Coder and CodeGemma families from 1.5B to 7B shows that performance holds at 3B and degrades only below that, which matters for the deployment story.

**Weaknesses:**

My main concern is that the ontology - which accounts for roughly 64 percentage points of SR in the ablation - is not described in enough detail to evaluate scalability. Section 4.1 covers the schema at a high level, and Appendix A.2 provides the class diagram (Figure 4) and one substitution example where the ontology detects that open_gripper should map to deactivate_vacuum (Figure 5, also described in the A.2 text). But how are the "substitutable-by" and "functionalEquivalent" relations actually authored? Manually for each robot pair? Through rules over API signatures? How many such relations exist for the four tested configurations, and what effort would be needed to add a new robot? The paper's Figure 4 Reasoner box mentions "deterministic rules + optional LM with effect-based validation" but this describes the inference procedure, not the construction of the underlying relations. Without this, it is hard to tell whether the approach generalizes or whether the tested scenarios are just narrow enough for manual specification to work.

The efficiency narrative has a gap. The paper defines an Inference Time (IT) metric in Appendix C.3 (Eq. 12) measuring wall-clock duration of LLM inference calls during error correction, but IT is relegated to the appendix breakdown tables (Tables 15-20) and absent from the main Table 1. The main-body efficiency story rests entirely on Grounding Overhead (GO), a token ratio that structurally favors localized editing - a method that patches 5 lines will always score better on GO than one that rewrites 50, whether or not the patches are correct or fast. For embodied control where latency matters, wall-clock time is what you actually care about, and it is odd that the authors formalized IT but chose not to report it.

All evaluation is in simulation (CoppeliaSim and Genesis). For a paper whose pitch is deployment on capacity-limited devices without cloud access, this is a real gap. The in-situ adaptation mechanism is particularly concerning here: the system patches executing code at runtime based on unit-test violations, but there is no discussion of what happens when a patch produces motions that violate joint limits, collide with obstacles, or are otherwise unsafe. No sandboxing, trajectory validation, or motion-safety layer is described. This matters less in simulation where you can just reset, but it becomes a serious issue on physical hardware.

There is also a baseline fairness question. RECENT's ontology provides structured substitution guidance that no baseline has access to - it tells the sLM exactly which API to replace with which (Appendix B confirms baselines receive reference skill code and API lists, not ontology mappings). The existing ablations do not cleanly isolate this advantage. The "Full Re-generation" row in Table 4 still uses the ontology for identifying mismatches, it just replaces FIM with full code regeneration (as described in Section 5.3). The "w/o Skill Ontology" row removes the ontology but also removes the structured editing that FIM relies on. What is missing is a baseline that performs FIM-based localized editing with some heuristic for finding edit locations (e.g., call-graph diffs or API name matching) but without ontology substitution hints. Finally, three seeds with no significance testing is thin - RepairAgent and RoboInspector both score 41.67% SR in Kinematic but with very different variance (RoboInspector at +/-7.53 vs RepairAgent at +/-28.04), and it is unclear which pairwise differences in Table 1 are reliable.

---

> ### Author Rebuttal · Authors · 2026-03-31
>
> We deeply appreciate the suggestions regarding presentation, which significantly strengthen our contribution.
>
> ---
>
> > **W1 Q1.**  Ontology construction and scalability.
>
> The ontology is not hand-authored for each task or robot pair. In our implementation, we define a shared ontology schema, adapted from prior robot knowledge representations [1], together with a fixed set of deterministic alignment rules, and annotate each robot once with its embodiment profile and primitive API metadata, including capabilities, parameters, and precondition/effect specifications. Based on these annotations, relations such as substitutable-by and functionalEquivalent are instantiated automatically through rule-based compatibility matching, rather than manually written as task-specific substitution facts. Across our experiments, the ontology remains compact, containing 5 robots, 40 primitives, 29 capabilities, 1,232 triples, and 40 instantiated substitution/equivalence relations.
>
> In practice, manual effort is limited to one-time schema alignment and specification of each robot's primitive APIs and capability annotations, using available resources such as URDF, SRDF, and skill specification files. Once these are defined, the ontology is constructed automatically by a deterministic Python pipeline in approximately 0.08 seconds, and adding new skills does not require manual edits to the ontology. We will clarify these construction details, ontology scale, and the expected extension effort for a new robot in the revised paper to make the scalability of the approach more explicit.
>
> > **W2 Q2.** Request to include IT in Table 1.
>
> We will include the Inference Time (IT) metric in the main Table 1, as detailed in `W4`.
>
> > **W3 Q4 Q6.** Safety safeguards and the role of in-situ adaptation.
>
> For safety, our ontology-based unit tests are not limited to API mismatch detection, but more generally validate embodiment consistency during execution. Under this view, unsafe patched behaviors can also be treated as validation failures within the same feedback loop. At the control level, this is complemented by robot-side safeguards, including built-in hardware safety limits and collision-aware planning with cuRobo [2], so execution is mediated by safeguarded low-level controllers within configured workspace boundaries.
>
> Extending to physical robots, low-level safety measures such as collision halting and joint-limit enforcement can be naturally incorporated as additional feedback signals in RECENT. We refer to our response to `fSaj W2` for real-world experiments with the Franka Research 3 and UR7e robots.
>
> In-situ adaptation is particularly beneficial in settings where the environment progresses continuously and execution errors propagate unless corrected, as it repairs localized mismatches during task execution. We also note that our simulation protocol does not assume reset-after-failure.
>
> > **W4 Q3 Q5.**  Baseline fairness and significance tests.
>
> We extend Table 4 to 5 seeds and add *w/ API name similarity*, a variant that retains FIM-based localized editing but removes ontology-guided substitution, using only heuristic API-name matching to identify edit locations and replacements.
>
> |Methods|SR↑|GO↓|EI↓|
> |-|-|-|-|
> |w/o Skill Ontology|21.67±7.45|2.35±1.55|2.57±0.59|
> |→ *w/ API name similarity*|*58.33±0.00*|*3.78±2.64*|*1.71±0.48*|
> |w/o In-situ Adaptation|71.67±4.56|1.77±1.02|3.05±5.12|
> |→ Full Re-generation|26.67±9.13|105.94±1.40|2.44±0.44|
> |RECENT|83.33±0.00|2.79±0.21|0.36±0.11|
>
> The added *w/ API name similarity* improves over w/o Skill Ontology but still underperforms RECENT by 25.00% in SR, showing that ontology-guided substitution provides benefits beyond localized FIM editing alone, especially for capability-level equivalences (e.g., `open_gripper` → `deactivate_vacuum`).
>
> We reproduced Table 1 with RECENT alongside the two most competitive sLM-based baselines on the same machine using 5 seeds (here, * denotes the privileged LLM-based baseline).
>
> |Method|SR↑|GC↑|GO↓|EI↓|IT↓|
> |-|-|-|-|-|-|
> |**Kinematic Variation**||||||
> |RoboInspector|41.00±4.59|60.83±6.20|179.40±81.75|1.15±0.45|59.98±33.97|
> |RepairAgent|42.00±4.11|64.05±2.50|54.17±19.87|1.73±0.14|50.53±17.35|
> |RECENT(ours)|73.00±3.71|81.76±1.07|7.85±3.00|0.72±0.35|1.77±0.90|
> |CAP-Codex*|67.50±8.48|80.45±4.81|84.29±25.78|1.23±0.51|7.99±4.11|
> |||||||
> |**End-effector Variation**||||||
> |RoboInspector|39.17±14.72|60.83±64.80|52.20±10.24|5.07±0.33|31.65±10.37|
> |RepairAgent|52.17±2.09|73.82±3.04|55.92±28.83|1.12±0.25|84.37±75.24|
> |RECENT(ours)|82.50±1.86|89.84±2.05|2.51±0.08|0.69±0.44|2.34±0.93|
> |CAP-Codex*|74.85±2.97|87.81±1.74|41.69±8.28|0.60±0.08|10.29±4.33|
>
> For SR, RECENT significantly outperformed all sLM-based baselines in both settings (Welch's t-test over 5 seeds, all p < 0.01) and achieved the lowest IT in both settings.
>
> [1] Representations for robot knowledge in the KnowRob framework
>
> [2] curobo: Parallelized collision-free minimum-jerk robot motion generation

---

> > ### Author Rebuttal · Reviewer_Tqg9 · 2026-04-03
> >
> > The ontology construction details (Q1) were the key missing piece. Automatic instantiation from one-time robot annotations (URDF/SRDF/API specs) via a deterministic pipeline is more practical than what I gathered from the paper. The scale (5 robots, 40 primitives, 40 substitution relations, 0.08s construction) makes scalability concrete.
> >
> > The API name similarity baseline (Q3) was exactly what I asked for and gives a clean decomposition: 21.67% without ontology or heuristics, 58.33% with FIM + name matching, 83.33% with full ontology. That 25-point gap confirms the value of capability-level equivalence mappings beyond simple string matching.
> >
> > 5-seed replication with Welch's t-test (p < 0.01) resolves the significance concern. IT now alongside GO - RECENT at 1.77s vs baselines at 50+ seconds - makes the efficiency case in wall-clock time, not just token ratio.

---

> > > ### Author Response · Authors · 2026-04-04
> > >
> > > Thank you for your thoughtful acknowledgement. We are very encouraged that our rebuttal has addressed your concerns, particularly regarding the ontology construction, baseline comparisons, and statistical validation. We sincerely appreciate your detailed feedback, which has helped us significantly strengthen the paper.
> > >
> > > Best regards,
> > >
> > > The Authors

---

### Official Review · Reviewer_piQB · 2026-03-12

**Soundness:** 3
**Presentation:** 3
**Significance:** 2
**Originality:** 2
**Overall Recommendation:** 4
**Confidence:** 3

**Summary:**

This paper studies the problem of skill grounding for embodied agents when deployment conditions differ from the environment where skills were originally developed. The authors propose RECENT, a refactoring-centric agent framework that represents skills as executable code and explicitly separates task semantics from embodiment-specific execution bindings. By performing code refactoring during deployment, the system adapts skills to new robot morphologies or environments using small language models (sLMs) instead of relying on large LLMs. Experiments on long-horizon manipulation tasks under kinematic and end-effector variations show that RECENT significantly improves task success and grounding efficiency while reducing token overhead and execution interruptions compared to several code-based and agent-based baselines.

**Compliance With Llm Reviewing Policy:**

Affirmed.

**Final Justification:**

My concerns are basically addressed, and I will raise my score to 4.

**Key Questions For Authors:**

-How robust is the refactoring process when the new embodiment requires structural changes to the skill logic (not just binding or parameter updates)?

-Could the authors provide more details on how the refactoring prompts and intermediate code representations are constructed for the sLM?

-How sensitive is the approach to the choice and size of the small language model used during grounding?

-Have the authors tested the method in real robotic deployments or with real sensor noise and actuation constraints?

**Limitations:**

yes

**Strengths And Weaknesses:**

Strengths

+The paper addresses a practical problem in embodied AI: adapting reusable skills across different robots or deployment contexts without retraining policies.

+The refactoring-based formulation is conceptually clean, explicitly separating semantic intent from execution details, which provides a clear mechanism for deployment-time grounding.

+Empirical results on long-horizon manipulation tasks show substantial improvements in success rate and grounding efficiency compared to several recent code-based baselines.

+The focus on small language models is relevant for real robotic systems where large LLM access may be limited or impractical.

Weaknesses

-The evaluation is relatively narrow in scope (primarily simulated manipulation tasks), making it unclear how well the method generalizes to broader embodied settings or real-world robots.

-The paper could better clarify the limits of the refactoring process, e.g., cases where structural changes to the skill logic are required rather than parameter or binding updates.

-Some implementation details of the refactoring pipeline and prompting strategy are briefly described, which may make exact reproduction difficult.

-Comparisons focus mainly on code-based approaches; it would be useful to discuss how the approach relates to policy-learning or modular skill-learning alternatives.

---

> ### Author Rebuttal · Authors · 2026-03-31
>
> We thank the reviewer for the insightful feedback and address the concerns below.
>
> > **W1 Q4.** Real-world test.
>
> To evaluate generalization beyond simulation, we conducted real-robot experiments to test RECENT.
>
> We evaluated RECENT in a retail 'scan-and-bag' scenario that requires coordinated manipulation skills for picking, contact-rich interaction, and target placement.
> The offline skill repository was built on the Franka Research 3 and deployed on the UR7e. We introduced unexpected obstacles into the scanner region to expose execution-time disturbances.
> This triggered a grasp-based recovery that was not viable on the UR7e due to the gripper’s limited aperture. RECENT instead adapted the skill code in situ to use a sweep primitive, allowing the task to complete with minimal interruptions.
>
> |Methods|# Successful Runs↑|GC↑|GO↓|EI↓|
> |-|-|-|-|-|
> |CaP|4/10|80.00±17.80|125.80±21.94|3.00±1.45|
> |RECENT|8/10|96.00±8.00|15.70±2.60|0.40±0.49|
>
> The final version will include hardware specifications, scenario configurations, and setup photos to improve reproducibility.
>
> > **W2 Q1.** Scope of refactoring process.
>
> To clarify the scope of refactoring, we classify structural changes in skill logic into two cases:
> (1) those resolvable within the existing skill ontology and
> (2) those requiring ontology-level capability extensions, which lie outside the scope of skill grounding and instead correspond to an incremental learning problem [1].
>
> For (1), the skill structure must be reorganized due to embodiment constraints while remaining expressible using existing ontology primitives. Such cases are handled through ontology-guided refactoring, where repeatedly failing code regions are restructured into finer-grained primitive sequences.
>
> (2) Ontology-level capability extension: When required capabilities are not represented in the ontology (e.g., transferring to a legged mobile manipulator requires navigation capabilities), these cases fall outside the scope of skill grounding. Once the ontology is extended, RECENT can operate on the extended ontology.
>
> [1] Incremental learning of retrievable skills for efficient continual task adaptation
>
> > **W3 Q2.** Implementation details.
>
> We show how determined conflicts (Sec. 4.2) are resolved with a concrete example. The full pipeline implementation is also available in our supplementary material including prompt details.
>
> Our prompt construction strategy begins with conflict localization.
> The ontology-based diagnosis identifies `open_gripper(env)` in the source skill as a conflict and derives the substitution target `deactivate_vacuum(env)`. From this conflict-substitution pair, we construct an intermediate code representation capturing the conflict span, context, and substitution hint in a structured format (e.g., `{"span": "open_gripper(env)", "context": (prefix, suffix), "hint": "deactivate_vacuum(env)", "metadata": {...}}`).
> This representation bridges the ontology output and the sLM input, and is then converted into the final FIM prompt:
>
> ```
> # Target API: deactivate_vacuum(..) -> releases the held object
> <|fim_prefix|>
>     move_to_position(..)
>     # Original code: open_gripper(..)
>     # MUST call: deactivate_vacuum(..)
> <|fim_suffix|>
>     move_parallel(..)
> <|fim_middle|>
> ```
>
> > **W4.** Comparison with learning-based approaches.
>
> We considered two learning-based baselines operating at deployment time: (1) a test-time training (TTT) approach [2], which updates additional modules using reference samples, and (2) a few-shot learning approach using an efficient prompt retriever (EPR) [3].
>
> Unlike RECENT, TTT and EPR require additional reference data at deployment time and incur substantial overhead from repeated updates.
>
> The table below reports the task success rate and grounding efficiency.
> |Methods|SR↑|GO↓|IGT↓|
> |-|-|-|-|
> |TTT (k=2)|57.72±0.80|61.17±3.93|46.06±0.50|
> |TTT (k=10)|66.16±8.59|75.83±6.86|64.24±0.15|
> |EPR (k=2)|59.72±2.41|49.30±18.84|12.86±0.15|
> |RECENT|81.94±2.41|2.57±0.03|1.98±0.03|
>
> We measure deployment-time efficiency using Initial Grounding Time (IGT), the time to produce an executable skill. RECENT achieves a 15.78% higher SR than TTT (k=10) while grounding 6.49x faster than EPR (1.98 s), where k denotes the number of learning samples.
>
> [2] The Surprising Effectiveness of Test-Time Training For Few-Shot Learning
>
> [3] Learning to Retrieve Prompts for In-Context Learning
>
> > **Q3.** Sensitivity to sLM choice and size.
>
> As shown in Table 2, RECENT shows low sensitivity to sLM choice, maintaining stable performance across model families and sizes.
>
> This robustness stems from RECENT's decomposed grounding process, which offloads mismatch diagnosis to ontology-based compatibility analysis and automatic localization while restricting repair to localized FIM-based code editing. This reduces the sLM's reasoning burden and helps maintain performance as complexity increases.

---

### Decision · Program_Chairs · 2026-04-30

**Decision:**

Accept (regular)

**Comment:**

This paper formulates the problem of robot code generation as a refactoring problem instead of generation from scratch; an easier task that is less error-prone for small LLMs than full code generation, and which leverages the skill decomposition and abstraction that programmatic representations naturally offer.

The reviews are in agreement that reformulating the robot code-generation as a refactoring problem to reuse previous solutions is a nice idea, and useful in practice. The main concerns are in the degree of innovation, and empirical evaluations. While programmatic representations and abstractions are fairly common now, the insight of refactoring for reuse is useful. The concerns about evaluation are also somewhat addressed by the additional results the authors shared in the response.

Overall, this paper seems like a decent systems contribution, with good empirical evaluations. The authors are urged to revise the final paper to incorporate the real robot results, and a clearer description of the contributions.